**METHOD**

# Haploflow: strain-resolved de novo assembly of viral genomes

Adrian Fritz[1,2], Andreas Bremges[1,2], Zhi-Luo Deng[1], Till Robin Lesker[1,2], Jasper Götting[2,3], Tina Ganzenmueller[2,3,4], Alexander Sczyrba[1,5], Alexander Dilthey[6,7], Frank Klawonn[8,9] and Alice Carolyn McHardy[1,2*]

\* Correspondence: alice.mchardy@
helmholtz-hzi.de
[1]Department of Computational
Biology of Infection Research,
Helmholtz Centre for Infection
Research, Braunschweig, Germany
[2]German Centre for Infection
Research (DZIF), Site
Hannover-Braunschweig,
Braunschweig, Germany
Full list of author information is
available at the end of the article

With viral infections, multiple related viral strains are often present due to coinfection or within-host evolution. We describe Haploflow, a deBruijn graph-based assembler for de novo genome assembly of viral strains from mixed sequence samples using a novel flow algorithm. We assess Haploflow across multiple benchmark data sets of increasing complexity, showing that Haploflow is faster and more accurate than viral haplotype assemblers and generic metagenome assemblers not aiming to reconstruct strains. We show Haploflow reconstructs viral strain genomes from patient HCMV samples and SARS-CoV-2 wastewater samples identical to clinical isolates.

Due to co-infection or within host evolution, in viral infections closely related strains, or haplotypes, might be present, with high average nucleotide identity (ANI) [1] to one another [2–5]. Modern sequencing technologies can capture this variation and computational assembly techniques reconstruct the individual genomes from the resulting data. Currently, there are predominantly two types of methods for this problem, viral haplotype assemblers [6, 7] and general (meta)genome assemblers [8–12]. Assembly of individual strains is very difficult, especially if variation is low and few reads span varying sites, resulting in highly fragmented strain genome reconstructions or consensus assemblies [13, 14].

(Meta)genome assemblers usually represent read data initially as a deBruijn (*k*-mer) graph and haplotype assemblers use string graphs [15–18]. String graphs, while being computationally more expensive to construct [19], due to overlap calculation for all read pairs, have the advantage of detecting mutations that co-occur on a single read [6], while for deBruijn graphs, this is limited to mutations occurring within the specified *k*-mer length [20]. String graphs are thus more sensitive in matching mutations to strains. If the strains have long stretches of identical sequences, co-occurrences may not happen, which typically is then solved by returning fragmented genome assemblies, where contigs are split between consecutive mutations that cannot be assigned to individual strains. As more contextual information is lost in the deBruijn graph, mutations appear as "bubbles" in the graph, where consecutive vertices are connected by more than one edge [21, 22]. (Meta)genome assemblers typically consider these bubbles as errors and follow different approaches for their resolution [22]. The popular SPAdes

assembler only considers one path of the bubble and thus loses the information of the second strain and reconstructs the dominant strain [9, 13, 23]. MEGAHIT instead terminates contigs prematurely if a bubble is encountered [8]. This leads to fragmented assemblies in the presence of closely related strains [13].

We here describe Haploflow, a new method and software for the de novo, strain-resolved assembly of viral genomes, which overcomes the problems for both types of methods, i.e., low speed versus loss of strain-specific information, by using information on differential coverage between strains to deconvolute the assembly graph into strain resolved genome assemblies. Haploflow thus does not require reads spanning multiple variable sites for strain resolved assembly of low divergent haplotype populations. As it is based on deBruijn graphs, it approaches the runtime behavior of modern metagenome assemblers. We demonstrate the ability of Haploflow to resolve strains fast and accurately on multiple data sets, including a low complexity HIV strain mixture to a complex, simulated virome sample consisting of 572 viruses with substantial strain-level variation, varying abundances and genome sizes as well as two data sets of clinical human cytomegalovirus (HCMV) and SARS-CoV-2 data.

## Results

We next describe the algorithm for creating and manipulating the assembly graph and the flow algorithm that gave Haploflow its name.

### deBruijn and unitig graph creation

The input to Haploflow is a sequence file including read sequences and specifying the *k*-mer size for constructing the deBruijn graph. Optionally, the lowest expected strain abundance (or *error rate*) can be specified, leading to removal of more rare *k*-mers from the graph, for graph simplification. Setting the *error-rate* size too low possibly makes the unitig graph and subsequent assembly more complex, while a too high value will prevent low abundant strains from being assembled.

First, a deBruijn graph is created from the reads, using ntHash [24] for *k*-mer hashing. Given the reads $R = \{r_1, ..., r_n\}$, a deBruijn graph $G = (V, E, k)$ contains all substrings of length *k-1* of $R$ as vertices $V$ and two vertices $u$ and $v$ are connected with a directed edge, if a substring of length $k$ exists, which has $u$ as prefix and $v$ as suffix [21]. In addition to this definition, in our deBruijn assembly graph, the count of every encountered *k*-mer in $R$ is stored for the respective edge. After creating this deBruijn assembly graph, all weakly connected components (called *CC*s, a set of vertices that are connected directly or indirectly to each other in the graph) of the graph are determined. The connected components are found with repeated depth-first searches, until every vertex has been visited and its connected component set. Afterwards, CCs are transformed individually into condensed versions of deBruijn graphs, so-called unitig graphs, where linear paths of vertices, having only one ingoing and one outgoing edge, are collapsed into one vertex.

This unitig graph has the following properties:

a) Every remaining vertex is a junction, having more than one ingoing or outgoing edge or being a source or sink. This means that all variation is found in vertices, all non-unique sequences (i.e., occurring in multiple haplotypes) are found in edges.

b) The unitig graph is a homeomorphic image of the input deBruijn graph, disregarding error correction. This means that no information is lost and the original deBruijn graph could be reconstructed.

When constructing this unitig graph, for each connected component, so-called junctions, vertices having a different in- from out-degree, or an in- or out-degree of more than one in the deBruijn graph are identified with a depth-first search. These will be the vertices of the new unitig graph, and their $k$-mers are maintained (Additional File 1: Fig. S1). The sequence of all the traversed $k$-mers is added to the connecting edge, and we define the length of an edge as the length of this sequence in base pairs. Starting from any junction, the next junction in the deBruijn graph is searched, passing vertices with exactly one ingoing and one outgoing edge until the next junction is found. Since all junctions are guaranteed to be searched and the transformation is deterministic, the choice of starting junction does not matter. When the next junction is found, the coverage of all the traversed edges is averaged and checked versus a threshold based on the *error rate* (Additional File 1: Fig. S1). If it is above, the target junction is also added as a vertex to the unitig graph and an edge with the (averaged) coverage value as the edges coverage is added between the two vertices. If the coverage is below the threshold, then neither the target vertex nor the edge is created and the next outgoing edge of the source is considered. This is repeated until all junctions have been searched, such that no vertices with in-degree = out-degree = 1 are remaining (Fig. S1). The resulting unitig graph is usually of drastically reduced size in comparison to the original graph, with sometimes less than 0.01% of vertices remaining. All linear paths of the original graph are condensed into single edges that represent stretches of unique contig sequences.

For every unitig graph a $k$-mer coverage histogram is built (Additional File 1: Fig. S1). These histograms reveal several key properties on our data sets: first, the coverage of reads belonging to one genome is approximately normally distributed around the "real" coverage of that genome [19, 20]. Second, if several sufficiently distinct (in terms of average nucleotide identity) genomes are present in a single unitig graph, all of them have a corresponding peak in the histogram. The longer a genome, the more different $k$-mers it includes, and accordingly, the higher the peak. If genomes are closely related, the peaks correspond to $k$-mers that are unique to the individual strains and additional, smaller peaks for common $k$-mers (across or within genomes).

Haploflow uses these coverage histograms as indication of the putative number of genomes [25] and their size relation as well as for error correction. Every read error will create $k$ erroneous $k$-mer vertices in the deBruijn graph [22, 26], with low coverage in comparison to the real coverage *cov* of the genomes. Since sequencing errors are rare in Illumina reads, most erroneous $k$-mers will only appear once [27, 28], with fewer $k$-mers appearing multiple times, creating an exponentially decreasing curve in the $k$-mer histogram. This information is factored into the error correction with too rare $k$-mers being removed (red line, Additional File 1: Fig. S1). The exact method and values used for error correction can be customized by the user, but by default, all $k$-mers with a coverage less than the first inflection point of the coverage histogram are filtered and every $k$-mer which has less than 2% of the coverage of its neighboring $k$-mers. This parameter can be increased when dealing with long read data to reflect the higher number of errors in current long read technologies.

## Assembly using the flow algorithm

In the second stage, the algorithm operates on the unitig graph. It infers and returns a set of contigs based on paths of similar coverages throughout the graph. The flow algorithm consists of three steps that are repeated until the whole graph has been resolved into contigs: (i) finding paths through the graph, (ii) assigning flow values to them, and (iii) determining the path sequence.

In the first step, the source vertex (with an in-degree of 0) with the highest coverage is selected from the unitig graph. Starting from this source, a modified Dijkstra's algorithm [29] is applied, which identifies the fattest path from a source to sink (a vertex having an out-degree of 0) based on edge coverages (Fig. 1, Fig. 2). The fatness of a path is defined by the minimal fatness of the edges on the path. The fatness of an edge is determined as the minimum of its coverage and the fatness of the path from the source until the current edge [30] and can also be called the "capacity" of the edge. The fattest path from a source to a sink is then determined by following the edges

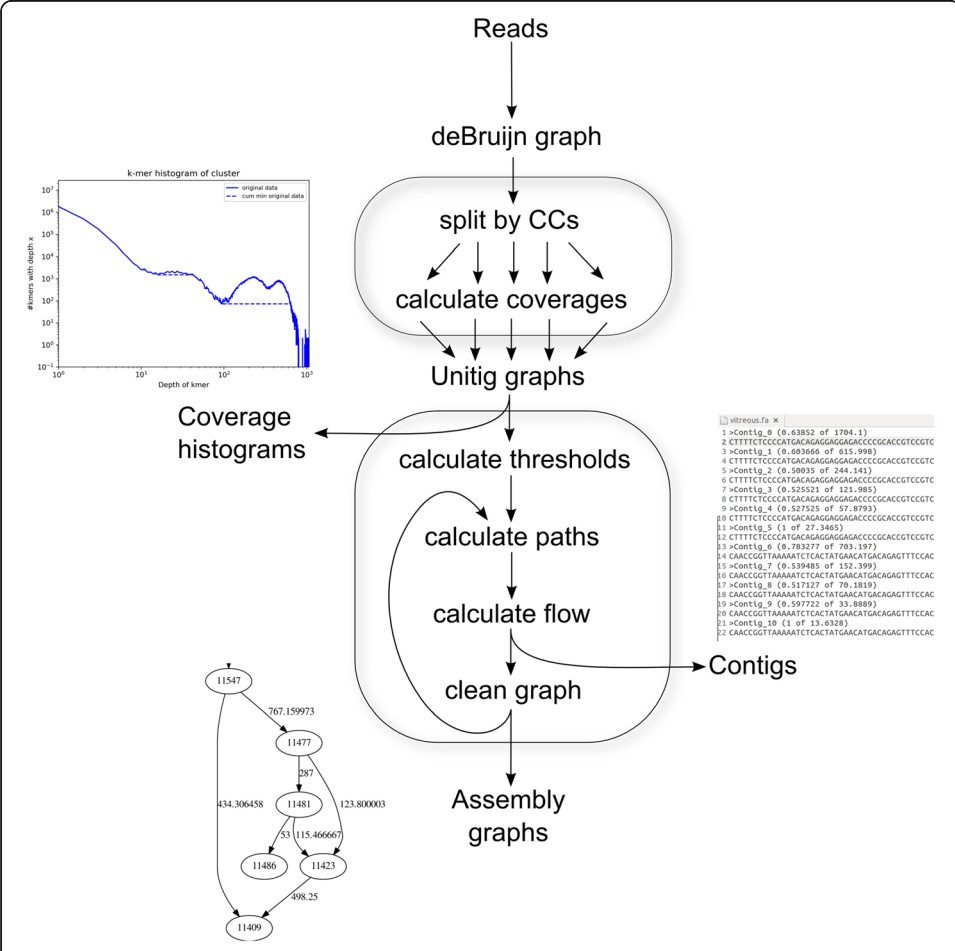

**Fig. 1** Flow chart of the Haploflow algorithm and its two parts: First, the construction of the deBruijn graph and operations thereon, namely splitting it by connected components and calculating coverages. Then, the creation of the unitig graphs per CC and the assembly process consisting of calculating the thresholds and the coverage histograms and the putative paths through the graphs. Next is the calculation of the concrete flows and thereby the generation of the contigs and finally the cleaning of the graph and the generation of the assembly graphs. As intermediate output, the assembly graph is created during every step (bottom left)

---

**Algorithm 1:** Fattest-path Dijkstra

**Data**: graph $G = (V, E)$ with edges $e = (v, w) \in E$, with $v, w \in V$, source $s \in V$, non-negative edge capacities $c(v, w)$ with $v, w \in V$

**Result**: Graph with edges labelled by their fatness, starting from source $s \in V$

1 **forall the** $v \in V - \{s\}$ **do**
2 $\quad$ $v.fat = 0$ $\hfill$ set fatness to 0
3 $\quad$ $s.dist = \infty$ $\hfill$ set distance to $\infty$
4 **end**
5 $Q = priority\_queue(V)$ $\hfill$ create priority queue keyed by $v.dist$
6 **while** $!(Q.isEmpty())$ **do**
7 $\quad$ $u = argmax(Q, fat)$ $\hfill$ select fattest vertex $u$ in $Q$
8 $\quad$ $del(u, Q)$ $\hfill$ remove $u$ from $Q$
9 $\quad$ **forall the** $v \in V$ *s.t.* $(u, v) \in E$ **do**
10 $\quad\quad$ $\hfill$ Breadth-first search through $G$
11 $\quad\quad$ **if** $v.fat < min\{u.fat, c(u, v)\}$ **then**
12 $\quad\quad\quad$ $v.fat = min\{u.fat, c(u, v)\}$ $\hfill$ update fatness
13 $\quad\quad\quad$ $v.dist = u.dist + length(e)$ $\hfill$ set $dist$ as distance to source
14 $\quad\quad\quad$ update $Q$ with new $dist$ values $\hfill$ update $Q$
15 $\quad\quad\quad$ $u.pred = v$ $\hfill$ set predecessor for backtracking
16 $\quad\quad$ **end**
17 $\quad$ **end**
18 **end**

---

**Fig. 2** The adapted Dijkstra algorithm used in Haploflow to find fattest paths through the unitig graph. Instead of determining the shortest paths from the source to all vertices, this algorithm determines the fattest path. The fatness is initialized as 0 for all vertices, but the source and then the graph is searched using a breadth-first search and based on the fact that the fattest path from a source $s$ to a sink $t$ is based on the edge with the lowest coverage along this path (lines 9 to 12)

maximizing fatness until the sink is found. All edges on this path are then marked with a path number. Subsequently, the coverage for all edges on this path is reduced by the path fatness, the next source is selected, and the previous steps are repeated until no edges with coverage remain.

Likely due to technical issues, such as amplification biases [31] and read errors [32], and biological structures such as genomic repeats [33], coverages do not follow a normal distribution globally and consequently some consecutive edges in the assembly graph may exhibit steep changes in coverage. This is the reason why Haploflow uses a two-step procedure for path finding: First, paths are found through the graph as described before. Instead of directly returning contigs for these paths, these paths are only putative, meaning that all paths and changes to the graph are temporary at first. The algorithm of Haploflow is then able to handle heterogeneous coverages across genomes, e.g., highly pronounced in amplicon data or sequence data with high error rates, by using the local, not global coverage distribution, and not absolute coverage, but relative coverage, i.e., the only assumption is that the ratio between haplotypes is somewhat conserved. Additionally, putative paths can get removed, if too many of its edges are already part of a previous putative path (Methods). If a path consists almost only of edges that have been used before, this is an indicator that these paths would lead to duplicated contigs. Finally, this results in a graph where all edges are marked with one or more paths they are assumed to be on.

In the second part of the path finding, we start again from the source with the highest coverage. Since we have all edges marked with the path that they are on, we can select the edge that is farthest away from our source on the same path and calculate the fattest path from the source to this sink. If Haploflow is not able to resolve the fatness unambiguously, for example because two outgoing edges have almost the same fatness, then the path is terminated in this vertex. This is to prevent formation of chimeric contigs, because locally two strains might have similar coverages. For the final path, a corresponding contig is returned and the coverage reduced permanently (see the "Methods" section). Then, all edges with capacity 0 and all vertices without any edges are removed and the flow algorithm starts anew from the source vertex. This procedure is repeated until the graph does not have any edges remaining.

Haploflow has multiple parameters that can be set to improve the assembly, if more information is given. If no additional information is given, Haploflow has default settings that usually already provide high quality assemblies. All the evaluations in this article were performed using these default parameters, i.e., a value for $k$ of 41, and an *error-rate* of 0.02. The value of $k = 41$ was chosen since too small (in comparison to read lengths) values for $k$ lead to more ambiguities and a higher $k$ might lead to fragmented assemblies. If $k$ does not exceed 50% of read-size, the assemblies are of comparable quality. The error-rate parameter was set to 0.02, because this is the value assumed to be the upper bound of errors in short-read sequencing [34] and can be increased when dealing with more error-prone reads like those from PacBio or Oxford Nanopore.

Additional parameters include a setting for detecting strains with very low absolute abundance (*strict*), for data sets with exactly two strains (*two-strain*), as well as an experimental mode for highly complex data sets with clusters containing five or more closely related strains.

### SARS-CoV-2 clinical and wastewater metagenome data

We reconstructed viral haplotypes using Haploflow from 17 clinical SARS-CoV-2 samples sampled in Northrhine-Westphalia, Germany (DUS, 5 Illumina short-read samples) and Madison, Wisconsin (WIS, 6 Illumina short-read and 6 Oxford Nanopore long-read samples). After correcting for PCR amplification and sequencing errors (see the "Methods" section), Haploflow identified two strains in nine samples, consistent with in-sample variation [35–37]. The assembled contigs were assessed with QUAST [38] using the Wuhan-Hu-1 isolate strain (RefSeq NC_045512.2) as reference genome. For all samples, Haploflow produced contigs spanning the complete genome, in 13 cases as a single contig. Haploflow reconstructed the consensus genome sequence(s) found in GISAID [39] with almost 100% identities as the major strain—from both Illumina and MinION data generated for all WIS samples (Fig. 3). For the Wisconsin strains, which were passaged for up to two rounds in cell cultures, the reconstructed minor strains from short read data had more evolutionary divergences. In comparison to calls from the variant caller Lofreq [44], (see the "Methods" section), which performs particularly well on mixed strain viral data [45], both identified 17 (65.4%) of overall 26 identified variant sites (mutations and up to 2 bp indels). Interestingly, most of these are C- > T transitions, indicating a tendency to alter genome composition [46]

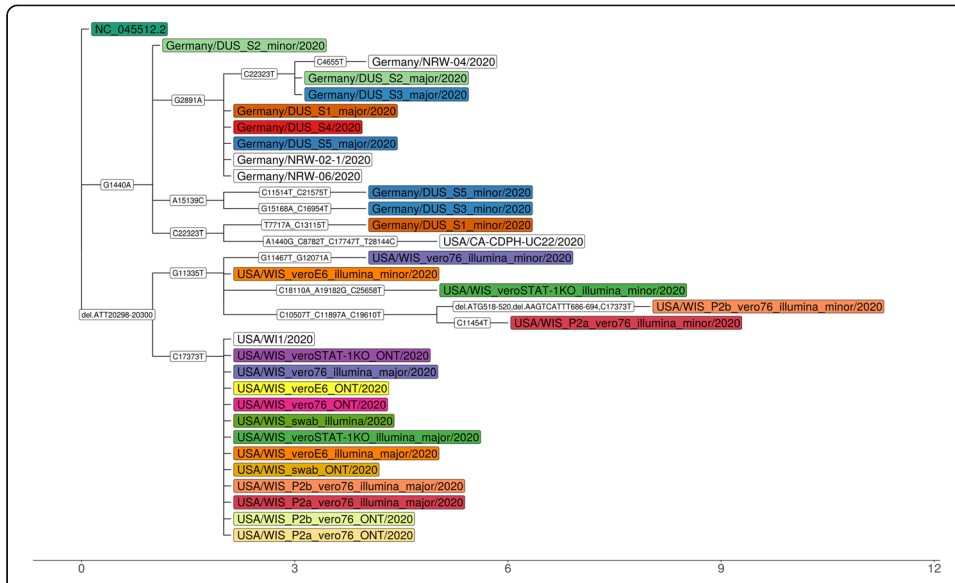

**Fig. 3** Phylogenetic relationships of reconstructed strain genomes inferred with Raxml [40, 41], including closely related (ANI greater than 99.99%, determined with MASH [42]) strains from GISAID [43]. Strains from the same sample are indicated by color, and "major" and "minor," based on their inferred abundances. Evolutionary events, including mutations, and indels are shown on edges

(Additional File 1: Table S1). In addition, Haploflow identified three longer deletions. Five (19.2%) "unique" LoFreq variants are located in error-prone regions (homopolymeric or strand biased) or at the very end of the genome. Four further low frequency sites (< 5%, 15.4%) were found by Haploflow and were also among low frequency Lofreq predictions.

In a study of eight shotgun metagenome samples of sewage from the San Francisco Bay Area [47], the authors manually assembled consensus SARS-CoV-2 genomes from seven samples and subsequently called variants with inStrain [48]. A comparison to common variants of clinical isolate genomes showed that most of the SNPs found in the data set could be detected in the isolate genomes, with the more (> 10%) abundant ones found in strains from California or the US. This and the abundance distribution of some SNPs over time suggested that the data set captured real genomic variation and that different SARS-CoV-2 strains were present in this data set. Haploflow with the option *strict 1* (reduced error correction threshold to account for shallow sequencing depth) and scaffolding, assembled full-length SARS-CoV-2 genomes for the same seven samples, recovering two strains for six of them (Additional File 1: Table S2). Strikingly, for all assemblies identical genomes of clinical SARS-CoV-2 isolates were identified in the GISAID database using minimap [49] v2.17 (Additional File 1: Table S2), mostly from samples obtained in the USA (5), and California (3), highlighting the ability of Haploflow to recover high quality, strain-resolved viral haplotype genomes from metagenomic data.

## Performance evaluation

We evaluated Haploflow on three simulated data sets with increasing complexity: a mixture of three HIV strains represented by error-free simulated reads, multiple in-vitro created mixtures with different proportions of two HCMV strains sequenced with

Illumina MiSeq [14, 50], and a simulated virome [51, 52] data set of 572 viruses, including 417 genomes in unique taxa and 155 genomes in common strain taxa with up to eleven closely related strains, to assess Haploflow's ability to assemble complex, larger data sets. Finally, we assembled HCMV genome data from clinical samples collected longitudinally over time from different patients [53], to characterize the within- and across patient genomic diversity of viral strains, including also larger genomic differences between individual strains in mixed-strain infections, which has not been possible so far. The evaluation was performed using metaQUAST [54] v.5.0.2, which is commonly used to evaluate metagenome assemblies and provides useful metrics for measuring completeness (genome fraction), continuity (NGA50, largest alignment), and accuracy (mismatches per 100 kb, duplication ratio) of assemblies and has specific options for analyzing strain-resolved assemblies. In addition, we calculated metrics for assessing strain-resolved assembly: the strain recall, specifying the fraction of correctly assembled strains (more than 90 (80)% genome fraction and less than 1 (5) mismatches/kb); the strain precision, specifying the fraction of correctly assembled strain genomes of all provided genome assemblies (true positives defined as in recall; total number of genome assemblies estimated as number of ground truth genomes with at least one mapping contig * duplication ratio); and the composite assembly quality score we previously defined [45]. This composite score takes six common assembly metrics (genome fraction, largest alignment, duplication ratio, mismatches per 100 kb, number of contigs and NGA50), normalizes them in the range of all results, such that $score(method) = \frac{value(method) - min(value(m \in methods))}{max(value(m \in methods)) - min(value(m \in methods))}$ for genome fraction, largest alignment and NGA50 and $score(method) = \frac{max(value(m \in methods)) - value(method)}{max(value(m \in methods)) - min(value(m \in methods))}$ for the other metrics, and then weighs with a weight of 0.3 for genome fraction and largest alignment, respectively, and a weight of 0.1 for the other metrics.

### HIV-3 in silico mixture

HIV, the human immunodeficiency virus, is a single-stranded RNA virus with an approximately 9.5 kb genome that infects humans, causing AIDS (acquired immunodeficiency syndrome). HIV evolves rapidly within the host and may also present as multi-strain infections [55, 56]. The three HIV-1 strains 89.6, HXB2 and JR-CSF, which are commonly used to evaluate viral haplotype assemblers [40, 57], were downloaded from NCBI RefSeq [41], mixed in the proportions 10:5:2 and error-free reads with a length of 150 bp and depth of 20,000 created with CAMISIM [42] and the wgsim read simulator [43]. These genomes differ mainly by SNPs and have an average nucleotide identity (ANI) of ∼ 95%. This threshold was chosen, because experiments on MEGAHIT and metaSPAdes showed that genomes more closely related than 95% will not be resolved [42].

   We benchmarked the quality of strain-resolved Haploflow assemblies for the three strain HIV data against five other de novo assemblers (SPAdes, metaSPAdes, megahit, PEHaplo, SAVAGE in de novo mode) with metaQUAST v.5.0.2, using multiple parameter settings, if defaults settings were undefined (QuasiRecomb [58], PEHaplo). Furthermore, we assessed five reference-based assemblers (GAEseq [59], SAVAGE ref-based mode, PredictHaplo, QuasiRecomb and CliqueSNV), which were provided with one strain genome for assembly.

Of all evaluated de novo assemblers, Haploflow performed best across all metrics and the composite assembly score (Additional File 1: Fig. S3, Table S3), assembling all three strains almost completely (more than 90%), with less than 1 mismatch/kb, providing no false positive strain assemblies—that for some methods (QuasiRecomb) reached several thousand strains—and with more than double the assembly contiguity (NGA50) than the second best method (PEHaplo). Haploflow was the only method assembling all strain genomes into complete contigs (Fig. 4A). Also in comparison to the reference-based assemblers, Haploflow performed best. SAVAGE in reference-based mode, run on a subsample of the data, performed similarly well in five of the eight metrics; however, it provided a substantially more fragmented assembly (lower NGA50, more contigs) and a strain genome with more mismatches. Haploflow also closely estimated the true underlying strain proportions, with predicted coverages of 10,371 for HIV 89.6, 5372 for HIV HXB2, and 1745 for HIV JR-CSF.

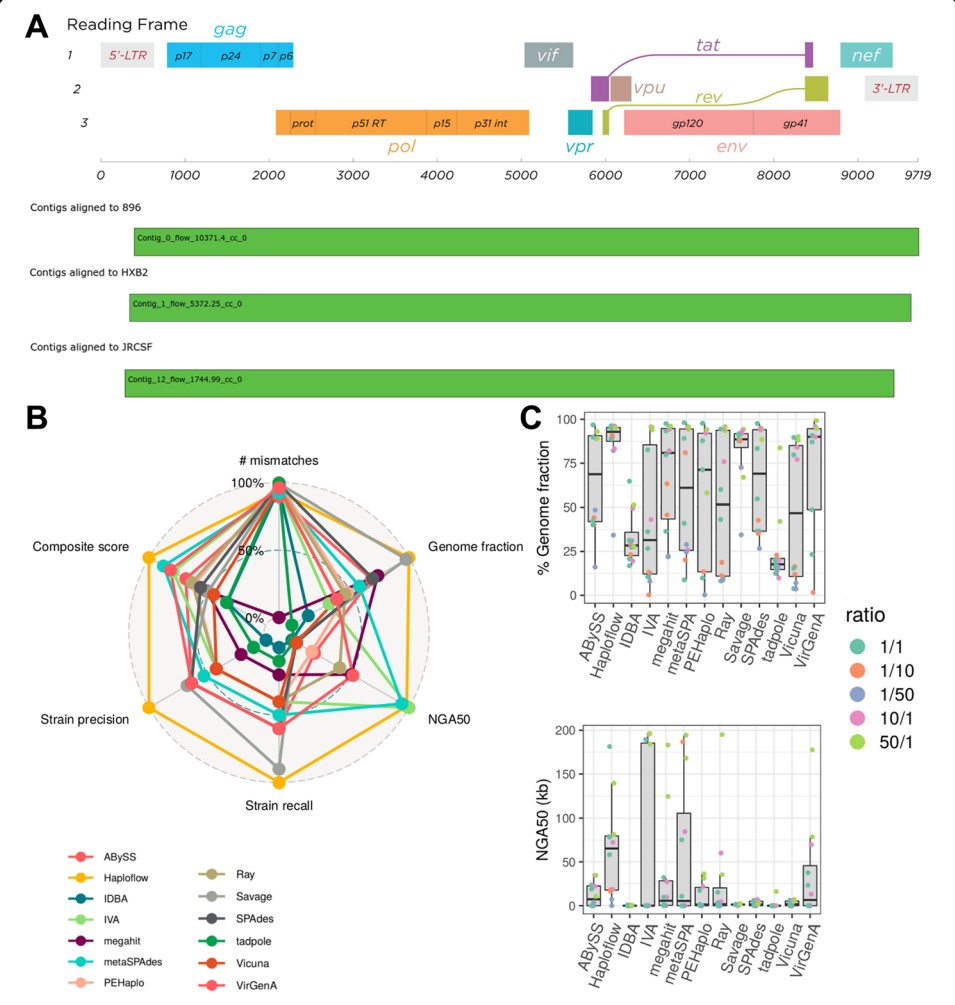

**Fig. 4 A** HIV genome structure [60] and Icarus plots [61] for three HIV strains reconstructed by Haploflow. For each of the three reference genomes, there is one contig spanning almost the complete genome. **B** Radar plot of relative performance with commonly used and strain-resolved genome assembly metrics for Haploflow and 12 other methods on the HCMV benchmark data (best values are at 100%, see the "Performance evaluation" section). Haploflow, in orange, ranks first in genome fraction, strain recall, strain precision, and composite score. **C** Boxplots with median and interquartile range of genome fraction and NGA50 values across samples for different methods

## HCMV in vitro mixtures

We next evaluated Haploflow on six lab-created mixtures of two HCMV strains sequenced with Illumina MiSeq as described previously [14]. HCMV is one of the largest human pathogenic viruses, causing severe illness in immunocompromised patients and infants, and possessing a double-stranded DNA genome of more than 220 kb [62]. The data set includes two different strain mixtures, denoted "TA" (strains TB40 and AD169, 97.9% ANI) and "TM" (strains TB40 and Merlin, 97.7% ANI), with three different mixture ratios each (1:1, 1:10, and 1:50), allowing us to test the ability of assemblers to resolve strains at varying abundances. We ran Haploflow on these data and compared the results to those of twelve other assemblers. These include nine (meta)genome assemblers (ABySS, IDBA, MEGAHIT, metaSPAdes, Ray, SPAdes, tadpole, IVA [63] and Vicuna [64]) also widely used for single-cell and virome data because of their accuracies and speed and three specialized viral haplotype assemblers delivering a result (reference-based SAVAGE, VirGenA [65] and PEHaplo). Four more viral haplotype assemblers, SAVAGE de novo, PredictHaplo [66], CliqueSNV [67], and QuasiRecomb [50], either did not return results [45] or were terminated after 10 days (Haploflow on average required 1.5 h per sample). Assemblies were evaluated using metaQUAST v.5.0.2 with the benchmarking workflow QuasiModo [45], based on common assembly metrics, the composite assembly score, recall and precision in strain-resolved genome assembly, as before, and the top performing methods falling in the 95–100% range of results identified for every metric.

Of the 12 evaluated de novo assemblers, Haploflow scored best in 5 of the 8 metrics, followed by metaSPAdes (best in 2 of 8: NGA50, duplication ratio), while PEHaplo, tadpole, IDBA, Vicuna, and IVA each scored best for one metric, respectively (Additional File 1: Table S4). Haploflow assemblies were of very high quality (Fig. 4B), recovering the most correct strain genomes (10 of 12), providing the best strain precision and composite assembly score (9.34 of 10), highest genome fraction (83.87%, Fig. 4C), and the most contiguous assemblies (NGA50 62,560, Fig. 4C). Interestingly, the similarly good NGA50 values of metaSPAdes and Haploflow were obtained in different ways, for the former due to a more contiguous assembly for the abundant strain (Additional File 1: Table S5), while only Haploflow and the haplotype assembler SAVAGE in reference-mode recovered more than 50% of the low abundant strain in several mixtures.

## Simulated virome data set

To test Haploflows' ability to recover viral strain genomes from complex data sets, we evaluated Haploflow, MEGAHIT, and metaSPAdes on the simulated virome data set from the Namib desert [51], which includes short-read data simulated from an in silico mixture of 572 viral genomes created to assess different assemblers [52]. It was not possible to run the *reference-free* haplotype-assemblers (SAVAGE, PEHaplo) on this data set. To assess the evolutionary divergence between the viral genomes, we identified clusters of similar genomes using dRep [68]. This resulted in 469 clusters in total, out of which 52 clusters had at least two (and up to 11, Additional File 1: Table S6) members with more than 95% ANI (average nucleotide identity), corresponding to 417 "unique" genomes and 155 genomes in common strain clusters. The 95% threshold was

chosen, since MEGAHIT and metaSPAdes were only able to resolve genomes less similar than that [42].

For the 155 common strain genomes, Haploflow correctly assembled 13-28.6% more sequence (62.85% genome fraction versus 55.58% and 48.88% for SPAdes and MEGA HIT, respectively). This was even more pronounced for clusters with genomes of at least eightfold coverage, for which 19.8–37.5% more genome sequence was correctly assembled (89.37% versus 74.58% and 64.99% for SPAdes and MEGAHIT, respectively). For the less abundant strains from these clusters, 32.7–45.3% more genome sequence was correctly assembled (87.37% versus 65.85% and 60.12% genome fraction, respectively). Even for the complete data set with "unique" genomes and low abundant genomes, Haploflow reconstructed genome fractions similar to the MEGAHIT and metaSPAdes assemblers (72.2 and 68.6% versus 66.6% genome fraction; Table S7), which performed best in the original publication.

### Analysis of clinical HCMV data

We used Haploflow with default parameters to reconstruct genomes from longitudinal clinical samples of eight HCMV positive patients, who had multi-strain infections [53] (Additional File 1: Table S8). QUAST was used to map Haploflow's contigs against the consensus strain of the first time point as reference genome, as the exact underlying strain genomes in the samples are unknown. Using the QUAST output, in particular the duplication ratio, the number of strains predicted by Haploflow was determined by rounding the duplication ratio and then clustering the contigs into that many clusters using Haploflow's predicted flow (using python's sklearn [69] *k*-means method). For each of the clusters, QUAST was re-run, again using the consensus as reference genome. Since the resulting genomes, in particular the low abundant (minor) strains, will inherently be different to the consensus to some degree, only the genome fraction is considered a relevant metric here. Additionally, to confirm that Haploflow created accurate strain-resolved contigs instead of consensus contigs, we compared clusters from the same patient at different time points with each other, finding that contigs from two clusters from consecutive time points showed ~ 99.9% ANI, while randomly matched clusters only had ~ 98% ANI.

For all patients with multi-strain infections, Haploflow reconstructed multiple complete genomes for at least one time point. For most patients, sequencing data of their infection exist for multiple time points and Haploflow recovered all strains, if the abundance of the lower abundant strain exceeded 6.8%, once as low as 6.1% (Additional File 1: Table S8). Haploflow reconstructed the genomes of both, or in two cases three, strains infecting the patient. Haploflow correctly predicted at least one lower abundant strain for 19 of 23 (82.6%) time points with multiple strain infections, correctly predicted 44 strains of the total 48 strains (91.7% recall) and only predicted (parts of) three unconfirmed, additional strains (93.2% precision). Haploflow also reproduced the results from the original publication [53], where a strain with a structurally altered genome established itself as dominant over two consecutive time points in patients SCTR1 and SCTR11, and also recovered three distinct strains from the SCTR18 sample (Additional File 1: Table S8). The samples for which Haploflow did not assemble a second strain had either a very low abundant second strain (4% for SCTR1-day91), a shallow

coverage (coverage of 22 and 38 for SCTR1-245 days and SCTR3-320 days) or a combination thereof (6.8% variant at 105 coverage for SCTR1-194 days).

Finally, we tested Haploflow on a HCMV sample for which the genotypes and proportion of both strains were previously determined by motif-matching [70] (Additional File 1: Fig. S4). Haploflow reconstructed both strains with a total of 19 contigs, 4 for the high abundant strain and 15 for the low abundant. The high abundant strain assembly matched the consensus strain with 0.87 mismatches/100 kb, an NGA50 value of 113,718 and 99.84% genome fraction. The contigs produced for the low abundant strain were also evaluated using the consensus sequence, showing 94.58% genome fraction and an NGA50 of 62,533. The largest contig showed a 7830 base sequence not present in the consensus sequence, but matching perfectly to another (BE/43/2011) HCMV sequence, demonstrating the ability of Haploflow to accurately phase different haplotypes.

### Runtime and memory consumption

Haploflow's run time depends on the three main steps (Fig. 1): first reading in the read data and building the deBruijn graph. For this, every read is split into $k$-mers, with a time complexity in $O(n)$ for the number of reads $n$. Since the maximal number of $k$-mers is constant in the number of reads and the length of the reads with $|k|$ = (length($n$) - $k$)·$|n|$, it is also in $O(k)$. Next the graph is split into CCs and the unitig graph constructed, with a time complexity of $O(k)$ using Tarjan's algorithm [71]. Finally, the overall complexity of the assembly step is dominated by finding the paths through the unitig graph. While theoretically there is an exponential number of different paths through a graph, every vertex can only be the source of a path once and every path has length at most $k$, since vertices cannot be visited multiple times on the same path. The worst-case complexity of the assembly step is thus in $O(k^2)$, where $k$ is the number of distinct $k$-mers. In practice, the number of paths is usually limited by the number of different strains, causing this step to also be linear time complexity.

For runtime assessment we compared Haploflow to SAVAGE and PEHaplo, the only other haplotype assemblers able to process the HCMV data, though SAVAGE only in *reference-based* mode, as well as metaSPAdes and MEGAHIT, which performed closest to Haploflow in terms of the summary score or is a very fast metagenome assembler, respectively (Table 1). On the HIV data, Haploflow was more than twice as fast than SAVAGE. The running time and memory requirements of Haploflow and metaSPAdes were comparable, while MEGAHIT was most efficient.

On the HIV three strain and the HCMV two strain mixtures, building the deBruijn graph and creation of the unitig graphs from the reads dominated the overall running time. For the HIV data, building the deBruijn and unitig graphs took ~ 8 min on a laptop with 4 cores and 16 GB RAM. The resulting single unitig graph included 281 vertices and assembly finished after 0.6 seconds. For the HCMV data, assembly on the same laptop required ~ 100 minutes, of which 85 were used for building the deBruijn and unitig graphs from the reads.

### Discussion and conclusions

Viral pathogens can evolve rapidly, leading to infections with multiple strains, by within-host evolution or multiple infections of the same host. Reconstructing their

**Table 1** Runtime and memory consumption of Haploflow, SAVAGE in de novo mode (version 0.4.1), metaSPAdes (3.14), and MEGAHIT (1.2.9). Time and memory is averaged for the HCMV mixtures. SAVAGE did not successfully complete on the HCMV in vitro mixtures and the simulated virome data. Values were calculated using Linux' time command

| Software/ data set | HIV 3 in silico mixture | | HCMV in vitro mixture | | Simulated virome | |
|---|---|---|---|---|---|---|
| Metric | CPU *user* time (seconds) | Memory peak (GB) | Avg. CPU user time (seconds) | Avg. memory peak (GB) | CPU user time (seconds) | Memory peak (GB) |
| Haploflow | 724 | **0.009** | 5170 | 17.509 | 18,245 | 47.678 |
| SAVAGE | 110,208 | 102.938 | 75,518 | 17.658 | – | – |
| PEHaplo | 10,127 | 11.819 | 58,920 | 13.998 | – | – |
| metaSPAdes | 1500 | 1.054 | 42,906 | 65.641 | 25,996 | 23.399 |
| MEGAHIT | **250** | 0.269 | **2910** | **0.754** | **9,690** | **2.148** |

genomes in a strain-resolved manner can substantially advance our understanding and capabilities to combat the diseases they cause. It is also key for genomic epidemiology, i.e., tracing viral spread using genomic information [72, 73] and genome-based viral phenotyping [74]. Strains can differ in their phenotypes, such as virulence, resistance, or the degree of their immune resistance to host immunity, which may be critical for the choice of therapy.

Strain-resolved de novo assembly from short-read as well as long read data generated in viral genome sequencing, however, is also extremely challenging. Haploflow fills a void between fast metagenome assemblers not aiming for strain-level resolution and viral haplotype assemblers for small viral genomes of a few kb in size. It combines the best of both worlds for strain-resolved genome assembly, by using the fast algorithms of the metagenome assemblers, i.e., deBruijn graph based assembly, together with a specialized flow algorithm for capturing strain variation, which allows to link variants that do not co-occur on reads.

Taken together, our results demonstrate a substantial performance improvement in strain-resolved assembly for Haploflow in comparison to sixteen other metagenome and viral haplotype assemblers evaluated across different benchmark data sets. The benchmark experiments on data sets with varying numbers of strains and abundances demonstrated that Haploflow can handle data sets with substantial variation in genomic coverage introduced by amplicon sequencing and resolved strains at different degrees of evolutionary divergences well, ranging from 95% ANI (HIV), over 98% ANI (HCMV), to more than 99% ANI (SARS-CoV-2 data). On the six lab-generated HCMV mixed strain data sets, Haploflow was top scoring in the most metrics (5 of 8) in comparison to twelve other assemblers. This performance improvement in strain recall, strain precision, composite score, genome fraction, and NGA50 was largely due to a better assembly of the less abundant strains. Except for Haploflow and SAVAGE, no method assembled low abundant strains to 50% on average and Haploflow had a *far* higher NGA50, creating long contigs rather than a highly fragmented assembly. On the clinical HCMV data tested, Haploflow almost perfectly (91.7% recall and 93.6% precision) assembled strains with variants predicted by variant callers and very closely predicted the abundances of second and third strains. On a three-strain HIV data set, Haploflow assembled all

three genomes almost entirely, with very few mismatches. This is reflected in Haploflow scoring top in all eight metrics, with a composite assembly score of 9.66 (out of 10), in comparison to 8.02 for the best *reference-based* assembler PredictHaplo, and of 6.28 for the best *reference-free* assembler PEHaplo.

Benchmarking on a rather complex simulated virome data set with 417 taxa with unique genomes and 155 genomes in common strain taxa showed that Haploflow successfully assembled 2-3 strains for "common strain taxa" with 2-11 strains, substantially better than the state-of-the-art metagenome assemblers and able to process the data set, other than the evaluated haplotype assemblers. This effect was particularly pronounced for strain genome coverages within a favorable (> 8) range for assembly. The abundance distribution of taxa in microbial communities is assumed to be oftentimes log-normal [13], with only a few abundant and a long tail of very low abundant ones with consequently low coverages. This indicates that Haploflow is suitable for processing many real world data sets and characterizing the more abundant strains, similar to the reference-based StrainPhlan strain-typing software [75]. Finally, Haploflow reconstructed multiple, full length SARS-CoV-2 strains from a multi-sample wastewater metagenome data set with exact matches to clinical isolate genomes found in the GISA ID database, highlighting the ability of Haploflow to recover high quality, strain-resolved viral haplotype genomes from metagenomic data.

In addition to short-read data, Haploflow also allows processing of long read data, which we demonstrated on the SARS-CoV-2 clinical data sets. For most applications dealing with low viral loads (e.g., the SARS-CoV-2 sequence data used here), PCR amplification is necessary to enrich viral reads. This naturally limits the possible maximum read length to the length of the PCR product, which is for those applications in the domain of short-read sequencing. The speed of the Haploflow algorithm principally also allows its extension to bacterial data, e.g., by adding multi-core and multi-$k$ support and modules for handling differently sized and structured microbial genomes. Thus, strain-resolved assembly from metagenome data for microbial taxa with several closely related strains could be a future application.

## Methods

### Exemplary clarification of path finding step realized in Haploflow

In the unitig graph, there are multiple paths between a source and a sink which (sans sequencing errors) correspond to the different strains present in a sample. The choice of the correct path follows the fatness algorithm described before. There is another factor though, namely the length of the fattest path, which Haploflow *also* maximizes. In Fig. S1 (Additional File 1), there is exactly one source, the vertex ACTA, and one sink, the vertex ATGC, but there are infinitely many paths from ACTA to ATGC, since CTAT to TCTA and TCTA to CTAT form a loop. To prevent this, Haploflow allows every edge only to be used *once* in every path finding step. This makes the particular loop in Fig. S1 "resolvable," the number of paths reduces to five:

$1 : ACTA \rightarrow CTAT \rightarrow TCTA \rightarrow CTAT \rightarrow ATGC$ with a fatness of 30
$2 : ACTA \rightarrow CTAT \rightarrow TCTA \rightarrow CTAC \rightarrow CTAT \rightarrow ATGC$ with a fatness of 45
$3 : ACTA \rightarrow CTAT \rightarrow ATGC$ with a fatness of 75
$4 : ACTA \rightarrow CTAC \rightarrow CTAT \rightarrow ATGC$ with a fatness of 25
$5 : ACTA \rightarrow CTAC \rightarrow CTAT \rightarrow TCTA \rightarrow CTAT$ with a fatness of 25

Just going by the fattest graph, path 3 would get selected, but this path is shorter than all other paths and thus only paths 2 and 5 can be selected, out of which path 2 has the higher fatness of 45 (the coverage of the first sequence). The next longest and fattest path is path 5 with a fatness of 25 (the coverage of the last sequence) and finally path 1 remains with a fatness of 30. Paths 3 and 4 do not exist at this point, since the capacity of all edges has been used.

### Algorithmic details of the flow algorithm

The fatness of a path is defined by the lowest fatness value of any edge along this path. Since the fatness of an edge might be underestimated if the coverage dropped for edges occurring before this edge in the path, it is not sufficient to just remove the calculated fatness when reducing flow along a path. Instead, the coverage of the source is set to 0 and for every other edge on the path the flow is reduced to *max(capacity - previously_ removed_flow, 0)* where *previously_removed_flow* is the flow removed from the last edge on the path. Since it is possible that edges are used multiple times, it is also possible that there are paths that have hardly any edges that are "unique" to that path. We call an edge *unique*, if it is part of exactly one path. If the fraction or length of unique edges of a path is too low, by default less than 500 bases, the path is removed for all edges on which it is not unique, to avoid overestimating the total number of paths in the graph. Edges with coverage of 0 will get removed, possibly producing new sources. If Haploflow crosses a junction with two or more outgoing edges with similar coverage values and cannot make an informed decision, which is the higher abundant path, Haploflow will break the contig at this position. This happens either if multiple strains have very similar coverages or on genomic repeats. The exact threshold for this break is derived using the *error_rate* and *strict/threshold* parameter: If the difference is less than the percentage value given or the (either explicitly stated or derived from *strict*) threshold, the contig is broken.

After the path has been found, the coverage of all unique edges on this path is reduced to 0, as no other path will be traversing this edge. If there is more than one path going over the edge, then the flow is reduced, corresponding to the expected coverage of the current edge. This value is the flow removed from the last visited unique edge, meaning that local increases and decreases in coverage are also captured. If the coverage of an edge would be reduced to 0, even though there are still paths going over this edge, the coverage is set to a dummy value such that it can still be used. On the other hand, if a path consists solely of non-unique edges, a duplication is assumed and the current path is not considered.

When permanently reducing the flow, it is not sufficient to remove the (overall) fatness of the path, since the fatness can only decrease (or stay the same) along a path, while the coverage values might fluctuate, based on amplification and sequencing strategy. To circumvent this, the flow is reduced by a "local fatness": all unique edges are removed as described before, for all other edges either flow removed from the last edge or, if the value is higher, of the average per-base removed flow, is taken as a baseline and depending on the fact whether the flow decreased or increased within the last edge, the flow to be removed is decreased and increased accordingly. If there would not be any flow remaining, a minimal value is left over.

## Reconstruction of full length SARS-CoV-2 sequences

In nine out of 17 SARS-CoV-2 samples and 6 out of 7 wastewater SARS-CoV-2 samples, QUAST reported a high duplication ratio for the Haploflow assembly; four out of five DUS and five out of twelve WIS samples. This can be explained by either artificially duplicating parts of the genome or the presence of two closely related strains. Since Haploflow did not construct single contig assemblies for all these strains, first a "scaffolding" step was performed: all contigs are clustered using $k$-means clustering on Haploflow's predicted abundance, the number of clusters depending on the duplication ratio. Then, using the NC_0455122.2 RefSeq strain, the contigs are extended to complete genomes, using the contigs bases for all parts of the genome covered by it. If a part of a genome is not covered by a strain, the bases from the highest abundant strain with bases at this position are inserted; if no strain has bases at this position, the reference base is inserted. If a position is covered twice, the base from the contig with higher flow is chosen. To reduce the number of false positive SNPs, an additional filtering step was performed to remove typical sequencing and PCR related artifacts (Additional File 1: Table S1), such as deletions within homopolymeric sites [34], mutations in short-tandem repeats [76], and mutations on sites with strong strand bias [77]. This for example removed a SNP included in the original submission of the DUS sample (in the HCMV data set) at position 4655 due to a high strand bias value.

Lofreq version 2.1.4 was run on the original reads and the variants filtered by an abundance value over 5% and a score of > 1000 to reduce the number of false positive calls. This filtered similar SNPs as the filtering of homopolymeric or strand biased sites performed for Haploflow.

## Supplementary information

---

**Additional File 1:.** Fig. S1-S4 and Tables S1-S8

**Additional file 2:.** Review history

---

### Acknowledgements

We thank Hadi Foroughmand, Sama Goliaei, Thorsten Klingen, Fernando Meyer, and Susanne Reimering for helpful discussions and Gary Robertson for technical support.

### Peer review information

### Review history

The review history is available as Additional file 2.

### Authors' contributions

A.C.M. conceived the research; A.C.M. and A.S. planned and coordinated the study; A.C.M., A.S., A.F., A.B., and F.K. designed the methodology; A.F. implemented the software. A.F., Z-L.D. and J.G. tested the software. T.R.L., Z-L.D., J.G., T.G., and A.D. provided data and analyzed results together with A.F., A.B., A.C.M., and F.K; A.F. and A.C.M. wrote the manuscript. All authors read and approved the manuscript.

### Funding

Funded by the Deutsche Forschungsgemeinschaft (DFG, German Research Foundation) under Germany's Excellence Strategy RESIST – EXC 2155 – Projektnummer 390874280, by the Volkswagen foundation, Communities Allied in Infection and by the DZIF (project number TI 12.002_00). Open Access funding enabled and organized by Projekt DEAL.

### Availability of data and materials

The code of Haploflow is available with a GPLv3 license on Github under https://github.com/hzi-bifo/Haploflow [78]. The version (v0.2) used for the assemblies in this publication is available under the DOI https://doi.org/10.5281/zenodo.4106497 [79]. All further scripts used are available on Github under https://github.com/hzi-bifo/Haploflow_

supplementary (DOI 10.5281/zenodo.4916177) [80]. All data sets and results from the performed evaluations are provided on Publisso with the DOI 10.4126/FRL01-006424451 [81].

## Declarations

### Ethics approval and consent to participate
Not applicable.

### Competing interests
The authors declare that they have no competing interests.

### Author details
[1]Department of Computational Biology of Infection Research, Helmholtz Centre for Infection Research, Braunschweig, Germany. [2]German Centre for Infection Research (DZIF), Site Hannover-Braunschweig, Braunschweig, Germany. [3]Institute of Virology, Hannover Medical School, Hannover, Germany. [4]Institute for Medical Virology, University Hospital Tuebingen, Tuebingen, Germany. [5]Faculty of Technology and Center for Biotechnology, Bielefeld University, Bielefeld, Germany. [6]Institute of Medical Microbiology and Hospital Hygiene, University Hospital, Heinrich-Heine-University Düsseldorf, Düsseldorf, Germany. [7]Genome Informatics Section, Computational and Statistical Genomics Branch, National Human Genome Research Institute, Bethesda, MD 20892, USA. [8]Department of Computer Science, Ostfalia University of Applied Sciences, Wolfenbuettel, Germany. [9]Biostatistics Group, Helmholtz Centre for Infection Research, Braunschweig, Germany.

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

## 