## [**Additional file 2:.** Review history · Genome Biology]

Review History

First round of review

Reviewer 1

Were you able to assess all statistics in the manuscript, including the appropriateness of statistical tests used? Yes. No additional statistics required.

Were you able to directly test the methods? No.

Comments to author:

The manuscript introduces a novel genome assembler that aims at strain-specific assembly of virus genomes. The solution approach is a combination of a de bruijn graph and a flow algorithm. Although many viral assemblies will be too complex for such software (presence of multiple, low abundant strains), it might be suitable for low complex viral genome and virome samples.

The manuscript clearly presents the software concept and the evaluation procedures. The three evaluation experiments are reasonably designed and their results are conclusive.

A major concern about this manuscript is the arbitrary selection of tools to compare to. Why have only metaSPADES, SAVAGE and MEGAHIT been used? I agree there are many more generic tools, which requires some selection. So, metaSPADES and MEGAHIT might be representative. However, there are more virus specific Haplotype assembler, which definitely should be considered: VirBin (Chen et al., 2019 BMC Bioinformatics), GAEseq (open source, preprint available) and more.

A further major concern is the focus on short reads. As long reads, such as PacBio or ONP, are particularly relevant for viral Haplotype assembly, their utility should be clearly discussed. Can Haploflow perform hybrid assemblies and incorporate long reads? How does it compare to metavirome hybrid assemblies of short and long reads with generic tools such as SPADES? How does it compare to long-read-only metaviromic assemblies?

Reviewer 2

Were you able to assess all statistics in the manuscript, including the appropriateness of statistical tests used? No.

Were you able to directly test the methods? Yes.

Comments to author:

Reconstructing viral haplotypes is an important and challenging problem. The authors of Haploflow delivered a new tool that achieved a better tradeoff between accuracy and speed than generic assembly tools and existing de novo haplotype reconstruction tools. The strengths of this work include: 1) speed. 2) can handle both RNA virus, DNA virus, and phages. 3) extensive tests on various types of data.

The speed part is well explained and justified because it is based on de bruijn graph rather than string graph. But it is not very clear to me about the accuracy of this method compared to other

haplotype reconstruction tools. The information used in the method, including graph pruning using node collapsing, search using abundance etc. is not new. They have been used by existing tools. So, the comparison in terms of the accuracy with other haplotype reconstruction tools should be more clear.

In particular, using coverage to distinguish different haplotypes has risks. Although the authors mentioned that the coverage of the same haplotype has normal distribution, it is not always true for real data. For example, if you look at the coverage for the first published SARS-CoV-2 by Zhang's group at Nature, the coverage is highly heterogeneous. It usually has multiple peaks. A similar problem was mentioned in a viral contig binning method that also used this coverage as the main feature to group assembled viral contigs (VIRBIN, A binning tool to reconstruct viral haplotypes from assembled contigs). Although it had some success, the heterogeneous coverage can still fail methods that rely on this feature. I think this is one of the main challenges for haplotype reconstruction tools. The authors need to explain more about the assumption of the normal distribution.

In the path finding algorithm based on the fattest path, did you consider the case that one edge can be further divided into sub-edges in different haplotypes? In that case, how the capacity of one edge is divided?

I appreciate the extensive experiments. But I suggest that the authors can add some information about the "purposes" of each experiment. This will make the experimental part easier to read. The first experiment is a little weak compared to others. Seems a more comprehensive experiment using simulated data should be done in order to deliver the following message: 1. How the number of strains affects Haploflow? 2. How the similarities of the strains affect Haploflow? (Viruses with different mutation rates can have very different overall sequence similarity) 3. How is the abundance/coverage distribution affect Haploflow? I think some of the questions are answered using other experiments. But the current organization makes it hard for readers to draw clear conclusions.

For HIV, there is a commonly used mock data set for evaluating haplotype reconstruction tool. It contains five strains with different abundance values. It is better to compare Haploflow with other cited tools such as SAVAGE, PEHaplo, and QuasiRecomb on that data set. In the first experiment, the authors also did not compare with these specially designed tools.

As the authors used many different tools in the experiments, it is thus hard to see the reasons. Some simple justifications should be given. For example, page 11, why use dRep and 95% ANI? 95% is quite low for viral haplotypes.

Minor:

Page 16, line 56. The complexity value has a box in the equation.

Reviewer 3

Were you able to assess all statistics in the manuscript, including the appropriateness of statistical tests used? No.

Were you able to directly test the methods? No.

Comments to author:

In the current paper the authors proposed Haploflow - a novel method for viral haplotype calling, aiming to de novo assemble the individual viral strains. The authors claim the method is reasonably fast and reconstruct both the haplotypes and their abundances accurately; such a method is of great interest for the field of epidemiology. However I have some major concerns about both the methods and the evaluation.

Method.

The method itself should be explained in much more details. It would help a reader if the authors recall the definition of the deBruijn graph, and explain better the construction of the unitig graph. In the current form it is not clear, what are the vertices and the edges of the unitig graph. The weights of the edges (the coverages) are also defined in a very dubious way.

In particular,

P4, 135: "all connected components (called CCs, a set of vertices that are connected directly or indirectly to each other in the graph)" -- it is not clear, do the authors mean strong or weak connected components;

P5, 139: "If there are multiple genomes present in a single unitig graph, then all of them will have a corresponding peak in the histogram." -- it is not clear why their coverages would not just summarize?

P5, 113: "Starting from a randomly chosen junction" -- does the result of the whole procedure depend on the choice of the junction here? The authors should prove that (or at least provide an explanation why) the proposed algorithm is correct.

P6, 126: "The fatness of a path is defined by the minimal fatness of the edges on the path. The fatness of an edge is determined as the minimum of its coverage and the fatness of the path from the source until the current edge". -- It looks like the fatness of an edge is not well-defined. Is the source unique? Is the path from the source until the current edge unique?

Based on this, I believe the paper would benefit from more formal explanation of the method (in particular, graph construction). Some illustrations (say, the construction the unitig graph from the deBruijn graph) could help a reader a lot.

Evaluation.

Haploflow has the input parameters -- the kmer size and (optional) the lowest expected strain abundance. The authors did not provide any discussion on how the performance of the method depends on the choice of these parameters, and how a user can choose them. While the lowest expected strain abundance can be estimated by a user from the sequencing depth and quality of the reads, it is absolutely unclear how a user can choose k.

Moreover, in the evaluation section the values of these parameters are given only for one dataset (HIV-3 in silico mixture).

P9, 159: We ran Haploflow on this data set with a kmer size of 41 and error rate of 0.02, which resulted in six contigs.

Was this parameter choice random? Did this choice maximize the assembly quality? What was the quality for other parameters values?

For the other datasets the authors did not provide the values of the parameters, which makes it problematic to reproduce the results.

P10, l22: "Haploflow also estimated the true proportions (10:5:2) very well, with predicted coverages of 10,371 for HIV 89.6, 5,372 for HIV HXB2 and 1,745 for HIV JR-CSF." --

While Haploflow reconstructed 6 contigs and there were only 3 strains, it is not clear how exactly it estimated the proportions so well.

The choice of the comparison metrics also should be discussed, and it would help a reader if the authors would use the same quality metrics for the performance analysis on all the datasets.

The comparison with other tools seems to be incomplete. In particular,

P10, l46: "Other viral haplotype assemblers, PEHaplo and QuasiRecomb, were either terminated after ten days"

In my experience, PEHaplo works pretty well and stable. Did the authors run it on the HIV-3 in silico mixture, or only on the HCMV in vitro mixtures?

P18, l34: "for which only SAVAGE, and only in reference-based mode, created an assembly"

While SAVAGE may provide very fragmented assemblies, there is a tool VG-flow from the same authors (<https://doi.org/10.1101/645721>) which improves SAVAGE assembly. I believe the comparison with its results is more relevant than the comparison with initial SAVAGE. Since it is possible to compare the Haploflow with the reference-based methods, I believe the paper would benefit from the comparison with such haplotype callers as PredictHaplo (10.1109/TCBB.2013.145) and CliqueSNV (<https://doi.org/10.1101/264242>).

Based on this, I believe that much more accurate and extensive comparison has to be done in the paper.

Minor comments:

- It is misleading that a plot on Figure S1 is called "histogram" several times.
- References to the figures are probably not valid (for example, p6 l26 (Alg. 1, Fig. 2) -> (Alg. 1, Fig. 1); p10 l5: (Fig 3A) -> (Fig 2A); p15 l20: (Fig. 4) ??).
- GitHub would benefit from a small working example.

Reviewer #1:

The manuscript introduces a novel genome assembler that aims at strain-specific assembly of virus genomes. The solution approach is a combination of a de bruin graph and a flow algorithm. Although many viral assemblies will be too complex for such software (presence of multiple, low abundant strains), it might be suitable for low complex viral genome and virome samples.

Benchmarking on a rather complex simulated virome data set (417 “unique” genome taxa and 155 genomes in common strain taxa) showed that Haploflow successfully assembled multiple strains for these “common strain taxa”, substantially better so than the state-of-the-art metagenome assemblers that we were able to run on these data (the reference-based haplotype assemblers and savage could not). This effect was particularly pronounced for strain genome coverages within a favorable (>8) range for genome assembly (**p. 17, Supplementary Table S5**).

To further demonstrate the value of Haploflow for the analysis of real metagenomic data with viral content, we added a new analysis for a metagenomic data set from sewage³, demonstrating the recovery of haplotype-resolved genomes from these data that were also recovered with identical sequences from clinical isolates. In the original study the authors manually assembled SARS-CoV-2 genomes and analysed single variant-based variation (“**Results**” section, **pages 10-12**):

“In a study of eight shotgun metagenome samples of sewage from the San Francisco Bay Area, the authors manually assembled consensus SARS-CoV-2 genomes from seven samples and subsequently called variants with inStrain⁹. A comparison to common variants of clinical isolate genomes showed that most of the SNPs found in the data set could be detected in the isolate genomes, with the more (>10%) abundant ones found in strains from California or the US. This and the abundance distribution of some SNPs over time suggested that the data set captured real genomic variation and that different SARS-CoV-2 strains were present in this data set. Haploflow with the option *strict 1* (reduced error correction threshold to account for shallow sequencing depth) and scaffolding (Supplement), assembled full-length SARS-CoV-2 genomes for the same seven samples, recovering two strains for six of them (Supplementary Table S8). Strikingly, for all assemblies identical genomes of clinical SARS-CoV-2 isolates were identified in the GISAID database using minimap¹⁰ v2.17 (Supplementary Table S8), mostly from samples obtained in the U.S. (5), and California (3), highlighting the ability of Haploflow to recover high quality, strain-resolved viral haplotype genomes from metagenomic data.”

Sample ID	# assembled strains by Haploflow	GISAID IDs of identical clinical isolate SARS-CoV-2 strains
Oakland 5/19	2	hCoV-19/USA/LA-SR0328/2020 hCoV-19/USA/CA-CSMC67/2020
Oakland 5/19 (2)	2	hCoV-19/Poland/PL_P31/2020 hCoV-19/Beijing/DT-BJ01/2020
Oakland 5/28	2	hCoV-19/USA/CA-CSMC25/2020 hCoV-19/USA/CA-CSMC67/2020
Oakland 6/09	1	hCoV-19/France/IDF-10064DR/2020
Oakland 6/30	2	hCoV-19/USA/WA-UW-11903/2020 hCoV-19/France/IDF-10064DR/2020
Oakland 6/30 (2)	2	hCoV-19/USA/CA-CSMC25/2020 hCoV-19/USA/LA-SR0328/2020
Marin 7/1	2	hCoV-19/USA/VA-DCLS-1271/2020 hCoV-19/USA/WA-UW-11903/2020

Suppl. Table S8: Number of SARS-Cov-2 genomes assembled by Haploflow from seven SARS-CoV-2 wastewater metagenome samples and GISAID¹¹ IDs of identical genomes recovered from clinical isolates. Strains are listed in order of their estimated abundances for individual samples.

The manuscript clearly presents the software concept and the evaluation procedures. The three evaluation experiments are reasonably designed and their results are conclusive.

We thank the reviewer for his or her positive assessment of our analysis.

A major concern about this manuscript is the arbitrary selection of tools to compare to. Why have only metaSPADES, SAVAGE and MEGAHIT been used? I agree there are many more generic tools, which requires some selection. So, metaSPADES and MEGAHIT might be representative. However, there are more virus specific Haplotype assembler, which definitely should be considered: VirBin (Chen et al., 2019 BMC Bioinformatics), GASeq (open source, preprint available) and more.

We agree with the reviewer that an extensive performance comparison against other methods is very important. Though in the first version of the manuscript we only evaluated three assemblers on the HIV data set, it included already an evaluation of ten assemblers, among these six viral haplotype-assemblers, on the HCMV benchmark data (**p. 14-15, Supplementary Table S2**). We selected these three on the HIV data set, because two of these are by far the most popular assemblers in the metagenomics domains and readers will likely be interested to see their performances, and also because metaSPAdes was the best performing assembler, second to Haploflow, on the HCMV data set.

To further address this comment, in the revised version we included results for a larger selection of methods on the HIV data set (**Supplementary Table S1, Suppl. Figure S3**); the reference-based viral haplotype assemblers (QuasiRecomb, CliqueSNV, GAEseq¹²), *de novo* haplotype assemblers (PEHaplo, SAVAGE *de novo*). To the benchmark on the HCMV data, we added evaluation of the viral haplotype assembler PEHaplo and VirGenA¹³ (another viral assembler), resulting in a total of 16 evaluated assemblers; twelve evaluated on the HCMV data (those optimized and capable of processing larger data sets) and nine on the HIV data (those optimized for haplotype-assembly). The method of GAEseq suggested by reviewer 1 sounds very interesting, but unfortunately GAEseq did not produce any results on the HIV data set, possibly due to the large number of reads in the data set (personal communication with author). The authors of GAEseq propose a more efficient software¹⁴, which unfortunately is not available yet. The second method suggested by reviewer 1, VirBin¹⁵, is a binner for already assembled viral contigs, which relies on high quality assemblies as input for binning, rather than producing them - so it was not possible to include it in the benchmark.

We summarized the new analysis in the updated “**performance evaluation**” sections in the text (**p. 12-15**), as follows:

“We evaluated Haploflow on three simulated data sets with increasing complexity: a mixture of three HIV strains represented by error-free simulated reads, multiple in-vitro created mixtures with different proportions of two HCMV strains sequenced with Illumina HiSeq, and a simulated virome data set of 572 viruses, with 417 genomes in unique taxa and 155 genomes in common strain taxa with up to eleven closely related strains, to assess Haploflow’s ability to assemble complex, larger data sets. Finally, we assembled HCMV genome data from clinical samples collected longitudinally over time from different patients, to characterize the within- and across patient genomic diversity of viral strains, including also larger genomic differences between individual strains in mixed-strain infections, which has not been possible so far. The evaluation was performed using metaQUAST v.5.0.2, which is commonly used to evaluate metagenome assemblies and provides useful metrics for measuring completeness (genome fraction), continuity (NGA50, largest alignment) and accuracy (mismatches per 100kb, duplication ratio) of assemblies and has specific options for analyzing strain-resolved assemblies. In addition, we calculated metrics for assessing strain-resolved assembly; the strain recall, specifying the fraction of correctly assembled strains (more than 90 (80)% genome fraction and less than 1 (5) mismatches/kb), the strain

precision, specifying the fraction of correctly assembled strain genomes of all provided genome assemblies (true positives defined as in recall; total number of genome assemblies estimated as number of ground truth genomes with at least one mapping contig * duplication ratio), as well as the composite assembly quality score, we previously defined. This composite score takes six common assembly metrics (genome fraction, largest alignment, duplication ratio, mismatches per 100 kb, number of contigs and NGA50), normalises them in the range of all results, such that $score(method) = \frac{value(method) - \min(value(m \in methods))}{\max(value(m \in methods)) - \min(value(m \in methods))}$ for genome fraction, largest alignment and NGA50 and $score(method) = \frac{\max(value(m \in methods)) - value(method)}{\max(value(m \in methods)) - \min(value(m \in methods))}$ for the other metrics and then weighs with a weight of 0.3 for genome fraction and largest alignment, respectively and a weight of 0.1 for the other metrics.”

“We benchmarked the quality of strain-resolved Haploflow assemblies for the three strain HIV data against five other *de novo* assemblers (SPAdes, metaSPAdes, megahit, PEHaplo, SAVAGE in *de novo* mode) with metaQUAST v.5.0.2, using multiple parameter settings if defaults settings were undefined (QuasiRecomb, PEHaplo). Furthermore, we assessed five reference-based assemblers (GAEseq, SAVAGE ref-based mode, PredictHaplo, QuasiRecomb and CliqueSNV), which were provided with one strain genome for assembly.

Of all evaluated *de novo* assemblers, Haploflow **performed best across all metrics and the composite assembly score** (Figure S3), assembling all three strains almost completely (more than 90%) and with less than 1 mismatch/kb of sequence, providing no false positive strain assemblies - that for some methods (QuasiRecomb) reached several thousand strains - with more than double the assembly contiguity (NGA50) than the second best method (PEHaplo). Haploflow was the only method assembling all strain genomes into complete contigs. Also in comparison to the reference-based assemblers, Haploflow performed best. SAVAGE in reference-based mode, run on a subsample of the data, performed similarly well in five of the eight metrics, however, provided a substantially more fragmented assembly (lower NGA50, more contigs) and a strain genome with more mismatches.”

“We next evaluated Haploflow on six lab-created mixtures of two HCMV strains sequenced with Illumina MiSeq. HCMV is one of the largest human pathogenic viruses, causing severe illness in immunocompromised patients and infants, and possessing a double stranded DNA genome of more than 220,000 base pairs¹⁷. This data set includes two different strain mixtures denoted “TA” (strains TB40 and AD169, 97.9% ANI) and “TM” (strains TB40 and Merlin, 97.7% ANI), with three different mixture ratios each - 1:1, 1:10 and 1:50, allowing us to test the ability of haplotype assemblers to resolve strains at varying abundances. We ran Haploflow on these data and compared the results to those of twelve other assemblers. These include nine (meta)genome assemblers (ABYSS¹⁸, IDBA¹⁹, MEGAHIT, metaSPAdes, Ray²⁰, SPAdes²¹, tadpole, IVA²² and Vicuna²³) also widely used for single-cell and virome data because of their accuracies and speed, and three specialised viral haplotype assemblers delivering a result (reference-based SAVAGE, VirGenA and PEHaplo). Four more viral

haplotype assemblers, SAVAGE *de novo*, PredictHaplo, CliqueSNV, QuasiRecomb, either did not return results or were terminated after ten days (Haploflow on average required 1.5h per sample). Assemblies were evaluated using metaQUAST²⁴ v.5.0.2 with the benchmarking workflow QuasiModo²⁵, based on common assembly metrics, the composite assembly score, recall and precision in strain-resolved genome assembly, as before, and the top performing methods falling in the 95-100% range of results identified for every metric.

Of the 12 evaluated *de novo* assemblers, Haploflow scored best in 5 of the 8 metrics, followed by metaSPAdes (best in 2 of 8; NGA50, duplication ratio), while PEHaplo, tadpole, IDBA, Vicuna and IVA each scored best for one metric, respectively (Supplementary Table S2). Haploflow assemblies were of very high quality, recovering the most correct strain genomes (10 of 12), providing the best strain precision and composite assembly score (9.34 of 10), highest genome fraction (83.87%) and the most contiguous assemblies (NGA50 62,560). Interestingly, the similarly good NGA50 values of metaSPAdes and Haploflow were obtained in different ways, for the former due to a more contiguous assembly for the abundant strain, while only Haploflow and the haplotype assembler SAVAGE in reference-mode recovered more than 50% of the low abundant strain in several mixtures.”

Suppl. Figure S3: Radar plot of relative performance for Haploflow and nine other methods on the HIV-3 *in silico* data set. Relative best performance is at 100%. Haploflow (dark blue) ranks first in *Strain recall*, *Strain precision* and *Composite score*.

GAEseq ^{a*}	-	-	-	-	-	-	-	-
---	---	---	---	---	---	---	---

Suppl. Table S2: Benchmark results for the HCMV data set. Shown are average values for the metaQUAST metrics over the six data sets and additional assembly metrics (see “performance evaluation”). For every metric, the best performing methods (95-100% range of results) are indicated. **Strain recall* includes correctly recovered genomes at two quality levels: more than 80(90)% genome fraction and less than 5(1) mismatches/kb. PEHaplo did not assemble one (TA-1-10) of the six mixtures.

	Strain recall*	Strain precision	Composite score	Genome fraction	Contigs	Mis-matches (100kb)	Duplication ratio	NGA50
Haploflow	10(3)/12	10/14	9.34	83.87± 10.37%	20.50±7. 20	166.26±1 22.97	1.20±0.1 5	62,560.42 ±35,233
metaSPAdes	5(4)/12	5/12	8.18	58.57±4. 44%	17.42±8. 38	184.13±3 37.14	1.01±0.0 1	60,008.25 ±37,089
SPAdes	6(4)/12	6/12	5.22	65.52±3. 94%	85.17±15 .75	40.38±29 .84	1.05±0.0 1	2,552.42 ±951
MEGAHIT	2(0)/12	2/23	4.71	68.01±8. 23%	324.83±2 44.29	2254.09± 1901.7	1.92±0.6 7	32,446.08 ±35,925
PEHaplo	4(3)/12	4/12	5.70	52.72± 18.07%	54.0± 78.17	13.04± 11.68	1.05± 0.07	10,960.1 ±6,602
tadpole	1(0)/12	1/12	3.15	24.47±13 .31%	39.92±12 .98	27.14±50 .67	1.00±0.0 0	1,344.3± 3,292
ABYSS	6(3)/12	6/12	6.41	64.88±4. 94%	20.92±8. 85	250.0±85 .39	1.05±0.0 1	12,399.25 ±5,157
Ray	4(4)/12	4/12	5.90	51.39±2. 27%	16.67±12 .53	67.54±63 .39	1.07±0.0 6	26,154.75 ±36,557
IDBA	0(0)/12	0/12	3.10	32.71±9. 52%	83.75±13 .24	104.46±7 6.39	1.03±0.0 1	154.67±1 78
Vicuna	4(1)/12	4/12	4.18	47.26±0. 93%	36.33±7. 79	104.05±7 1.31	1.02±0.0 1	2,657.67 ±704
IVA	4(3)/12	4/12	7.42	43.23±16 .2%	11.92±8. 22	121.61±1 85.89	1.02±0.0 3	63,773.0 ±49,449
VirGenA	6(5)/12	6/12	7.63	47.25±1. 35%	5.67±2.4 8	102.41±1 07.01	1.01±0.0 0	33,324.58 ±30,049
SAVAGE	9(5)/12	9/17	4.61	82.43±15 .69%	283.17±9 8.89	33.86±28 .37	1.46±0.1 4	1,245.33 ± 349

Figure 3: A: HI virus genome structure²⁷ and Icarus plots²⁸ for three HIV strain genome assemblies by Haploflow. For each reference strain, one assembled contig spans almost the complete genome. **B:** Radar plot of relative performance with commonly used and strain-resolved genome assembly metrics for Haploflow and 12 other methods on the HCMV benchmark data (best values are at 100%, see Performance evaluation). Haploflow, in orange, ranks first in genome fraction, Strain recall, Strain precision and Composite score. **C:** Boxplots with median and interquartile range of genome fraction and NGA50 values across samples for different methods.

A further major concern is the focus on short reads. As long reads, such as PacBio or ONP, are particularly relevant for viral Haplotype assembly, their utility should be

clearly discussed. Can Haploflow perform hybrid assemblies and incorporate long reads? How does it compare to metavirome hybrid assemblies of short and long reads with generic tools such as SPADES?

We have clarified in the **discussion (p. 20-22)** that “Haploflow does allow processing of long read data which we demonstrated on the SARS-CoV-2 clinical data sets. For most applications dealing with low viral loads (e.g. the SARS-CoV-2 sequencing demonstrated in this article), PCR amplification is necessary to enrich viral reads. This naturally limits the possible maximum read length to the length of the PCR product, which is for those applications in the domain of short-read sequencing.

For analysis of long read data, we recommend to increase the error rate parameter:

- **Page 10:** “The error-rate parameter was set to 0.02, because this is the value which is assumed to be the upper bound of errors in short-read sequencing and can be increased when dealing with more error-prone reads like those from PacBio or Oxford Nanopore”

How does it compare to metavirome hybrid assemblies of short and long reads with generic tools such as SPADES?

SPAdes in hybrid mode, hybridSPAdes, just like the other SPAdes derivatives does not perform strain-resolved assembly, but creates “consensus assemblies”, therefore it cannot be used for this application. Because of the comparably high error rate or low coverages obtained with long read sequencing technologies, long reads are either used to scaffold existing (short-read) assemblies (hybridSPAdes²⁹, Unicycler³⁰) or are extensively polished using short read assemblies (Metaflye³¹ + FreeBayes³²). Therefore, these methods could profit from using a more strain-resolved assembly as input, like those that Haploflow provides.

How does it compare to long-read-only metaviromic assemblies?

There are currently only very few methods for long read only assembly, e.g. Metaflye and Canu³³, which do not provide strain-resolved assemblies though. Furthermore - to the best of our knowledge - there are no real data sets for long-read metaviromics with strain variation. Therefore, this question at the moment seems largely of academic interest, though may become more relevant in the future, once the throughput of long read technologies and viral enrichment protocols are further developed.

Reviewer #2:

Reconstructing viral haplotypes is an important and challenging problem. The authors of Haploflow delivered a new tool that achieved a better tradeoff between accuracy and speed than generic assembly tools and existing de novo haplotype reconstruction tools. The strengths of this work include: 1) speed. 2) can handle both RNA virus, DNA virus, and phages. 3) extensive tests on various types of data. The speed part is well explained and justified because it is based on de bruijn graph rather than string graph. But it is not very clear to me about the accuracy of this method compared to other haplotype reconstruction tools. The information used in the method, including graph pruning using node collapsing, search using abundance etc. is not new. They have been used by existing tools. So, the comparison in terms of the accuracy with other haplotype reconstruction tools should be more clear.

In response we have added further extensive comparisons with other methods (**Supplementary Tables S1, S2**), as well as additional plots (**Fig 2, S3**), metric calculation (**p. 12-13**), using the most recent version of the benchmarking software metaQUAST v.5.0.2, optimized for assessing strain-resolved assemblies. We substantially extended the discussion of the assembly performances to further clarify that Haploflow does provide a substantial improvement of assembly quality, in comparison to 16 other tested methods. We updated the **performance evaluation HIV-3 in silico (p. 13-14)** and **HCMV in vitro (p. 14-15)** sections accordingly:

“We benchmarked the quality of strain-resolved Haploflow assemblies for the three strain HIV data against five other *de novo* assemblers (SPAdes, metaSPAdes, megahit, PEHaplo, SAVAGE in *de novo* mode) with metaQUAST v.5.0.2, using multiple parameter settings if defaults settings were undefined (QuasiRecomb, PEHaplo). Furthermore, we assessed five reference-based assemblers (GAEseq, SAVAGE ref-based mode, PredictHaplo, QuasiRecomb and CliqueSNV), which were provided with one strain genome for assembly.

Of all evaluated *de novo* assemblers, Haploflow **performed best across all metrics and the composite assembly score** (Figure S3), assembling all three strains almost completely (more than 90%) and with less than 1 mismatch/kb of sequence, providing no false positive strain assemblies - that for some methods (QuasiRecomb) reached several thousand strains - with more than double the assembly contiguity (NGA50) than the second best method (PEHaplo). Haploflow was the only method assembling all strain genomes into complete contigs. Also in comparison to the reference-based assemblers, Haploflow performed best. SAVAGE in reference-based mode, run on a subsample of the data, performed similarly well in five of the eight metrics, however, provided a substantially more fragmented assembly (lower NGA50, more contigs) and a strain genome with more mismatches.”

Suppl. Figure S3: Radar plot of relative performance for Haploflow and nine other methods on the HIV-3 *in silico* data set. Relative best performance is at 100%. Haploflow (dark blue) ranks first in *Strain recall*, *Strain precision* and *Composite score*.

“We next evaluated Haploflow on six lab-created mixtures of two HCMV strains sequenced with Illumina MiSeq¹⁶. HCMV is one of the largest human pathogenic viruses, causing severe illness in immunocompromised patients and infants, and possessing a double stranded DNA genome of more than 220,000 base pairs¹⁷. This data set includes two different strain mixtures denoted “TA” (strains TB40 and AD169, 97.9% ANI) and “TM” (strains TB40 and Merlin, 97.7% ANI), with three different mixture ratios each - 1:1, 1:10 and 1:50, allowing us to test the ability of haplotype assemblers to resolve strains at varying abundances. We ran Haploflow on these data and compared the results to those of twelve other assemblers. These include nine (meta)genome assemblers (ABYSS¹⁸, IDBA¹⁹, MEGAHIT, metaSPAdes, Ray²⁰, SPAdes²¹, tadpole, IVA²² and Vicuna²³) also widely used for single-cell and virome data because of their accuracies and speed, and three specialised viral haplotype assemblers delivering a result (reference-based SAVAGE, VirGenA and PEHaplo). Four more viral haplotype assemblers, SAVAGE *de novo*, PredictHaplo, CliqueSNV, QuasiRecomb, either did not return results or were terminated after ten days (Haploflow on average required 1.5h per sample). Assemblies were evaluated using metaQUAST²⁴ v.5.0.2 with the benchmarking workflow QuasiModo²⁵, based on common assembly metrics, the composite assembly score, recall and precision in strain-resolved genome assembly, as before, and the top performing methods falling in the 95-100% range of results identified for every metric.

Of the 12 evaluated de novo assemblers, Haploflow scored best in 5 of the 8 metrics, followed by metaSPAdes (best in 2 of 8; NGA50, duplication ratio), while PEHaplo, tadpole, IDBA, Vicuna and IVA each scored best for one metric, respectively (Supplementary Table S2). Haploflow assemblies were of very high quality, recovering the most correct strain genomes (10 of 12), providing the best strain precision and composite assembly score (9.34 of 10), highest genome fraction (83.87%) and the most contiguous assemblies (NGA50 62,560). Interestingly, the similarly good NGA50 values of metaSPAdes and Haploflow were obtained in different ways, for the former due to a more contiguous assembly for the abundant strain, while only Haploflow and the haplotype assembler SAVAGE in reference-mode recovered more than 50% of the low abundant strain in several mixtures.”

Figure 3: A: HI virus genome structure²⁷ and Icarus plots²⁸ for three HIV strain genome assemblies by Haploflow. For each reference strain, one assembled contig spans almost the complete genome. **B:** Radar plot of relative performance with commonly used and strain-resolved genome assembly metrics for Haploflow and 12 other methods on the HCMV benchmark data (best values are at 100%, see Performance evaluation). Haploflow, in orange, ranks first in genome fraction, Strain recall, Strain precision and Composite score. **C:** Boxplots with median and interquartile range of genome fraction and NGA50 values across samples for different methods.

In particular, using coverage to distinguish different haplotypes has risks. Although the authors mentioned that the coverage of the same haplotype has normal distribution, it is not always true for real data. For example, if you look at the coverage for the first published SARS-CoV-2 by Zhang's group at Nature, the coverage is highly heterogenous. It usually has multiple peaks. A similar problem was mentioned in a viral contig binning method that also used this coverage as the main feature to group assembled viral contigs (VIRBIN, A binning tool to reconstruct viral haplotypes from assembled contigs). Although it had some success, the heterogeneous coverage can still fail methods that rely on this feature. I think this is one of the main challenges for haplotype reconstruction tools. The authors need to explain more about the assumption of the normal distribution.

The reviewer is correct about this - and we have improved our description of the respective passages to further clarify. The algorithm of Haploflow is able to handle heterogeneous coverages across genomes, by using the local, not global coverage distribution, and not absolute coverage, but relative coverage, i.e. the only assumption is that the ratio between haplotypes is somewhat conserved. In particular the SARS-CoV-2 data had significant amplification bias which Haploflow was able to amend this way. This is described in more detail now:

- **Page 6:** “Haploflow uses these coverage histograms as indication of the putative number of genomes and their size relation as well as for error correction”
- **Page 7:** “Due to technical issues, such as amplification biases and read errors, and biological structures such as genomic repeats, coverages across a genome do not follow a normal distribution globally and consequently some consecutive edges in the assembly graph may exhibit steep changes in coverage”
- **Page 7:** “The algorithm of Haploflow is able to handle heterogeneous coverages across genomes, e.g. highly pronounced in amplicon data or sequence data with high error rates, by using the local, not global coverage distribution, and not absolute coverage, but relative coverage, i.e. the only assumption is that the ratio between haplotypes is somewhat conserved.”

In the path finding algorithm based on the fattest path, did you consider the case that one edge can be further divided into sub-edges in different haplotypes? In that case, how the capacity of one edge is divided?

This question has prompted us to add a figure (**Figure S1**) explaining the deBruijn graph and unitig graph structure in more detail. There are two cases, either the k -mers of two haplotypes coincide (i.e. their capacity is added) and they appear as a single edge or - in case they differ in that k -mer - they form “bubbles” in the graph with the capacity equal to the coverage of the haplotype, i.e. two vertices connected by multiple edges.

Figure S1: The deBruijn graph (1) and its corresponding Unitig graph (2) for three related sequences and their coverage (3). The red k -mers and edges between them are part of linear paths and are replaced by a single red edge in the unitig graph. The edges are labelled with the “capacity”, the sum of the coverages of the sequences going over them, in the deBruijn graph and the average capacity of all smoothed edges in the Unitig graph - which in this case is the same as the original capacity. Some of the edges represent one (capacities 25, 30, 45), some two (capacity $70 = 45 + 25$) and some all (capacity $100 = 45 + 30 + 25$) of the sequences.

I appreciate the extensive experiments. But I suggest that the authors can add some information about the "purposes" of each experiment. This will make the experimental part easier to read. The first experiment is a little weak compared to others. Seems a more comprehensive experiment using simulated data should be done in order to deliver the following message: 1. How the number of strains affects Haploflow? 2. How the similarities of the strains affect Haploflow? (Viruses with different mutation rates can have very different overall sequence similarity) 3. How is the abundance/coverage distribution affect Haploflow? I think some of the questions are answered using other experiments. But the current organization makes it hard for readers to draw clear conclusions.

Thank you for this suggestion. We re-ordered the experiments and added an explanation why these data sets were chosen on **pages 13, 14 and 17**, respectively, and a paragraph to the **discussion (p. 21-22)** providing answers to questions 1-3:

- **Page 13:** “The three HIV-1 strains 89.6, HXB2 and JR-CSF, which are commonly used to evaluate viral haplotype assemblers [...]. These genomes differ mainly by SNPs and have an average nucleotide identity (ANI) of ~95%. This ANI threshold was chosen, because experiments on MEGAHIT and metaSPAdes showed that genomes more closely related than 95% will not be resolved.”
- **Page 14:** “[...] allowing us to test the ability of assemblers to resolve strains at varying abundances”
- **Page 17:** “To test Haploflows ability to recover viral strain genomes from complex data sets, we evaluated Haploflow on the simulated virome data set from the Namib desert”

“Benchmarking on a rather complex simulated virome data set with 417 taxa with unique genomes and 155 genomes in common strain taxa showed that Haploflow successfully assembled 2-3 strains for these “common strain taxa”, with 2-11 strains, substantially better so than the state-of-the-art metagenome assemblers able to process these data, that the haplotype assemblers (PEHaplo and SAVAGE) could not. This effect was particularly pronounced for strain genome coverages within a favorable (>8) range for assembly. The abundance distribution of taxa in microbial communities is assumed to oftentimes follow a log-normal distribution⁴, with only a few abundant and a long tail of very low abundant ones with consequently low coverages, which indicates that this performance is suitable for many real world data sets, similar to the approach realized in the reference-based StrainPhlan strain-typing software³⁴. The benchmark experiments on data sets with varying numbers of strains and abundances demonstrated that Haploflow performs best for strains being present at different, rather than at the same abundances. It can handle data sets with substantial variation in genomic coverage introduced by amplicon sequencing and resolved strains at different degrees of evolutionary divergences well, ranging from 95% ANI (HIV), over 98% ANI (HCMV), to more than 99% ANI (SARS-CoV-2 data).”

For HIV, there is a commonly used mock data set for evaluating haplotype reconstruction tool. It contains five strains with different abundance values. It is better to compare Haploflow with other cited tools such as SAVAGE, PEHaplo, and QuasiRecomb on that data set. In the first experiment, the authors also did not compare with these specially designed tools.

The HIV-3 data set contains three out of the five strains commonly used for evaluating the other Haplotype assemblers and we added the Haplotype assemblers SAVAGE, PEHaplo, QuasiRecomb, GAEseq and CliqueSNV to the evaluation.

The HIV-5 mixture which both SAVAGE and PEHaplo ran on had relatively long insert sizes (600bp) for their read-overlap strategy. Since Haploflow is not dependent on co-occurring variants, it is able to use the more realistic shorter reads (e.g. 2x150bp from NovaSeq) and still produce accurate results. Both SAVAGE and PEHaplo did not report duplication ratios in their evaluation, because these tools did indeed assemble the strains being present, but also many more “strains” (**Table S1**) which are more chimeric sequences between strains. Both these tools produced far more than the ~50kb of sequence expected for five strains with 10kb each. This is why we chose to evaluate Haloflow on the shown HIV-3 data set.

As the authors used many different tools in the experiments, it is thus hard to see the reasons. Some simple justifications should be given. For example, page 11, why use dRep and 95% ANI? 95% is quite low for viral haplotypes.

We extended our evaluation and showed the performance of all tools - if applicable - over all datasets now (**Figure 3, Tables S1, S2**), clarifying choices in the text. The reason that not all tools are included in all analyses, e.g. for the HCMV data set and the simulated virome, is that it was not possible to run most of the haplotype assemblers on them. Thus, for HCMV we only can show results for SAVAGE ref-based and PEHaplo, and for the simulated virome we could not retrieve assemblies for methods other than the metagenome assemblers. We also added an explanation for the 95% ANI threshold, which has been shown as a threshold of genome similarity for which MEGAHIT and metaSPAdes stop producing strain-resolved genomes but instead produce “consensus contigs”³⁵:

- **Page 13:** “This threshold was chosen because experiments on MEGAHIT and metaSPAdes showed that genomes more closely related than 95% will not be resolved”
- **Page 17:** “The 95% threshold was chosen since MEGAHIT and metaSPAdes are only able to resolve genomes less similar than that”

Minor:

Page 16, line 56. The complexity value has a box in the equation.

Corrected, thank you.

Reviewer #3:

In the current paper the authors proposed Haploflow - a novel method for viral haplotype calling, aiming to de novo assemble the individual viral strains. The authors claim the method is reasonably fast and reconstruct both the haplotypes and their abundances accurately; such a method is of great interest for the field of epidemiology. However I have some major concerns about both the methods and the evaluation.

Method.

The method itself should be explained in much more details. It would help a reader if the authors recall the definition of the deBruijn graph, and explain better the construction of the unitig graph. In the current form it is not clear, what are the vertices and the edges of the unitig graph. The weights of the edges (the coverages) are also defined in a very dubious way.

We added a definition and a figure explaining the deBruijn graph and unitig graph construction and explained the definitions more clearly (**p. 4, Figure S1, Suppl. Material**).

- **Page 4:** “Given the reads $R = \{r_1, \dots, r_n\}$, the deBruijn graph $G = (V, E, k)$ contains all substrings of length k of R as vertices V and two vertices u and v are connected with an edge if the prefix of u overlaps with the suffix of length $k - 1$ of v or vice versa³⁶, i.e. $(u, v) \in E \Leftrightarrow u_{1\dots k-1} = v_{2\dots k} \vee u_{2\dots k} = v_{1\dots k-1}$.”
- **Suppl. Material:** Added a paragraph describing the path finding algorithm in more detail (**p. 32**)

Figure S1: The deBruijn graph (1) and its corresponding Unitig graph (2) for three related sequences and their coverage (3). The red k -mers and edges between them are part of linear paths and are replaced by a single red edge in the unitig graph. The edges are labelled with the “capacity”, the sum of the coverages of the sequences going over them, in the deBruijn graph and the average capacity of all smoothed edges in the Unitig graph - which in this case is the same as the original capacity. Some of the edges represent one (capacities 25, 30, 45), some two (capacity $70 = 45 + 25$) and some all (capacity $100 = 45 + 30 + 25$) of the sequences.

In particular,

P4, l35: "all connected components (called CCs, a set of vertices that are connected directly or indirectly to each other in the graph)" -- it is not clear, do the authors mean strong or weak connected components;

We clarified that “weakly connected components” are meant.

P5, l39: "If there are multiple genomes present in a single unitig graph, then all of them will have a corresponding peak in the histogram." -- it is not clear why their coverages would not just summarize?

All k-mers for which two genomes differ will show in the histogram and produce a peak specific for the unique k-mers. Since these k-mers have to exist (otherwise the two strains would be the same), some peaks will show. The overlapping k-mers will produce another peak, but since Haploflow does not use the global k-mer distribution for resolving haplotypes, this does not hinder the assembly. We added a sentence explaining this on **p. 5** and **p. 6**.

- **Page 5:** "If multiple sufficiently distinct (in terms of average nucleotide identity) genomes are present [...]"
- **Page 6:** "If genomes are very closely related, then the peaks will consist of *k*-mers which are unique to the individual strains and there will be another peak for the common *k*-mers."

P5, l13: "Starting from a randomly chosen junction" -- does the result of the whole procedure depend on the choice of the junction here? The authors should prove that (or at least provide an explanation why) the proposed algorithm is correct.

The output does not depend on the chosen junction since the unitig graph can be uniquely defined for a given deBruijn graph as the homeomorphic image of the deBruijn graph where all vertices with exactly one ingoing and one outgoing edge have been smoothed away. We added an explanation why our algorithm provides this transformation (**p. 5**).

- **Page 5:** "[...] passing vertices with exactly one ingoing and one outgoing edge until the next junction is found. Since all junctions are guaranteed to be searched and the transformation is deterministic, the choice of starting junction does not matter."
- **Page 5:** "This is repeated until all junctions have been searched, such that no vertices with in-degree = out-degree = 1 are remaining (**Figure S1**)"

P6, l26: "The fatness of a path is defined by the minimal fatness of the edges on the path. The fatness of an edge is determined as the minimum of its coverage and the fatness of the path from the source until the current edge". -- It looks like the fatness of an edge is not well-defined. Is the source unique? Is the path from the source until the current edge unique?

We added a more formal description of "fatness" and clearly distinguish between "coverage", "capacity" and "fatness" (**p. 6-7**).

- **Pages 6-7:** "The fatness of a path is defined by the minimal fatness of the edges on the path. The fatness of an edge is determined as the minimum of its coverage and the fatness of the path from the source until the current edge and can also be called the

“capacity” of the edge. The fattest path from a source to a sink is then determined by following edges maximising fatness until the sink is found.”

The source is unique in the sense that the source is “the edge with the highest capacity” (as stated in the paper, **p. 6**). Haploflow does not require the paths to be unique - they are not necessarily as described on the same page - and we describe on **page 7** how Haploflow overcomes these issues.

- **Page 7:** “Likely due to technical issues, such as amplification biases and read errors, and biological structures such as genomic repeats, coverages do not follow a normal distribution globally and some path edges may exhibit steep changes in coverage, which causes downstream fatness values to be skewed. This is the reason why Haploflow uses a two-step procedure for path finding: First, paths are found through the graph as described before. But instead of directly returning contigs for these paths, these paths are only putative, meaning that all paths and changes to the graph are temporary first”

Based on this, I believe the paper would benefit from more formal explanation of the method (in particular, graph construction). Some illustrations (say, the construction the unitig graph from the deBruijn graph) could help a reader a lot.

We added more formal description of parts of the algorithm to increase understanding (**p. 4, 6-7**) and added a figure explaining the deBruijn graph and the unitig graph as well as the transformation (**Figure S1**). Finally, we added additional details about the method in the **Supplementary Material** (“*Exemplary clarification of path finding step realized in Haploflow*”).

Evaluation.

Haploflow has the input parameters -- the kmer size and (optional) the lowest expected strain abundance. The authors did not provide any discussion on how the performance of the method depends on the choice of these parameters, and how a user can choose them. While the lowest expected strain abundance can be estimated by a user from the sequencing depth and quality of the reads, it is absolutely unclear how a user can choose k.

Haploflow recommends a default setting of $k=41$, which works well in most cases. We added a sentence about how parameters can alternatively be chosen (**p. 10**) as well as an explanation that all data sets were evaluated using Haploflow’s default parameters and what these are (**p. 10**).

- **Page 10:** “If no additional information is given, Haploflow has default settings that usually already provide high quality assemblies. All the evaluations in this article were performed using these default parameters, i.e. a value for k of 41, and an *error-rate* of 0.02. The value of $k = 41$ was chosen since too small (in comparison to read

lengths) values for k lead to more ambiguities and a higher k might lead to fragmented assemblies. If k does not exceed 50% of read-size, the assemblies are of comparable quality. The error-rate parameter was set to 0.02, because this is the value assumed to be the upper bound of errors in short-read sequencing and can be increased when dealing with more error-prone reads like those from PacBio or Oxford Nanopore. Additional parameters include a setting for detecting strains with very low absolute abundance (*strict*), for data sets with exactly two strains (*two-strain*), as well as an experimental mode for highly complex data sets with clusters containing five or more closely related strains.”

Moreover, in the evaluation section the values of these parameters are given only for one dataset (HIV-3 in silico mixture).

P9, l59: We ran Haploflow on this data set with a kmer size of 41 and error rate of 0.02, which resulted in six contigs.

Was this parameter choice random? Did this choice maximize the assembly quality? What was the quality for other parameters values?

These parameters are Haploflow’s default parameter set and we did not do any parameter optimisation. We added this information on **p. 10**. A value of k between 30 and 50 has been proven to work best in most cases^{37,38}. Assembly quality was comparable for different parameter sets (e.g. higher k , higher error rate).

For the other datasets the authors did not provide the values of the parameters, which makes it problematic to reproduce the results.

We used default parameters for all data sets and now also state this (**p. 10**).

P10, l22: "Haploflow also estimated the true proportions (10:5:2) very well, with predicted coverages of 10,371 for HIV 89.6, 5,372 for HIV HXB2 and 1,745 for HIV JR-CSF." --

While Haploflow reconstructed 6 contigs and there were only 3 strains, it is not clear how exactly it estimated the proportions so well.

Haploflow reports the estimated coverage for each contig individually. In the previous version (without filtering contigs shorter than 500bp) of Haploflow, it had produced six contigs and for each of the genomes there was a contig basically spanning the entire genome (>9000 base pairs and three <500bp) which we based our previous proportion analysis on. In the latest Haploflow version we filtered the three short contigs and the analysis was based on the remaining three contigs for the three strains, which have the proportions as described (now on **p. 14**).

The choice of the comparison metrics also should be discussed, and it would help a reader if the authors would use the same quality metrics for the performance analysis on all the datasets.

As suggested, we added further explanations of the metrics used, as outlined below. For evaluation on the benchmark datasets, we used very commonly used metrics for metagenome assembly evaluations (e.g. CAMI I challenge), as described in the metaQUAST publication, a composite assembly score defined previously and now added two more intuitive metrics, strain precision and strain recall, specific to the problem. For the simulated virome, these metrics per genome are hard to compare and for SARS-CoV2 the real genomes are not known, so for these data sets we focus on the genome fraction. We changed the **performance evaluation** section accordingly (p. 12-13):

“We evaluated Haploflow on three simulated data sets with increasing complexity: a mixture of three HIV strains represented by error-free simulated reads, multiple in-vitro created mixtures with different proportions of two HCMV strains sequenced with Illumina HiSeq, and a simulated virome data set of 572 viruses, with 417 genomes in unique taxa and 155 genomes in common strain taxa with up to eleven closely related strains, to assess Haploflow’s ability to assemble complex, larger data sets. Finally, we assembled HCMV genome data from clinical samples collected longitudinally over time from different patients, to characterize the within- and across patient genomic diversity of viral strains, including also larger genomic differences between individual strains in mixed-strain infections, which has not been possible so far. The evaluation was performed using metaQUAST v.5.0.2, which is commonly used to evaluate metagenome assemblies and provides useful metrics for measuring completeness (genome fraction), continuity (NGA50, largest alignment) and accuracy (mismatches per 100kb, duplication ratio) of assemblies and has specific options for analyzing strain-resolved assemblies. In addition, we calculated metrics for assessing strain-resolved assembly; the strain recall, specifying the fraction of correctly assembled strains (more than 90 (80)% genome fraction and less than 1 (5) mismatches/kb), the strain precision, specifying the fraction of correctly assembled strain genomes of all provided genome assemblies (true positives defined as in recall; total number of genome assemblies estimated as number of ground truth genomes with at least one mapping contig * duplication ratio), as well as the composite assembly quality score, we previously defined. This composite score takes six common assembly metrics (genome fraction, largest alignment, duplication ratio, mismatches per 100 kb, number of contigs and NGA50), normalises them in the range of all results, such that $score(method) = \frac{value(method) - \min(value(m \in methods))}{\max(value(m \in methods)) - \min(value(m \in methods))}$ for genome fraction, largest alignment and NGA50 and $score(method) = \frac{\max(value(m \in methods)) - value(method)}{\max(value(m \in methods)) - \min(value(m \in methods))}$ for the other metrics and then weighs with a weight of 0.3 for genome fraction and largest alignment, respectively and a weight of 0.1 for the other metrics.”

The comparison with other tools seems to be incomplete. In particular, P10, 146: "Other viral haplotype assemblers, PEHaplo and QuasiRecomb, were either terminated after ten days"

In my experience, PEHaplo works pretty well and stable. Did the authors run it on the HIV-3 in silico mixture, or only on the HCMV in vitro mixtures?

In the version we used first, PEHaplo did indeed not produce any contigs. With the latest version, we were able to run PEHaplo on both the HIV-3 and the HCMV data sets and added it to the evaluation. Additionally, we added more methods (QuasiRecomb, CliqueSNV and PredictHaplo) to the HIV-3 evaluation (**p. 13-14, Supplementary Table S1**), where Haploflow clearly outperforms these. With the exception of PEHaplo, all of the newly added methods did not produce contigs for the HCMV data set. PEHaplo is part of the HCMV evaluation now and is ranked 7th overall (Haploflow 1st, SAVAGE 10th, **Table S2**) among 13 methods.

P18. l34: "for which only SAVAGE, and only in reference-based mode, created an assembly"

While SAVAGE may provide very fragmented assemblies, there is a tool VG-flow from the same authors (<https://doi.org/10.1101/645721>) which improves SAVAGE assembly. I believe the comparison with its results is more relevant than the comparison with initial SAVAGE.

VG-flow³⁹, can be understood as a scaffolder, as it relies on a given assembly and we did not compare Haploflow to scaffolding tools. In any case, if the initial assembly is missing haplotypes or is overestimating the number haplotypes, VG-flow does not correct these effects, which is why we chose to evaluate Haploflow only against SAVAGE.

Since it is possible to compare the Haploflow with the reference-based methods, I believe the paper would benefit from the comparison with such haplotype callers as PredictHaplo (10.1109/TCBB.2013.145) and CliqueSNV (<https://doi.org/10.1101/264242>).

We added a comparison with PredictHaplo, CliqueSNV and QuasiRecomb as reference-based tools on the HIV-3 data set (**p. 13-14, Supplementary Table S1**). We always used one of the genomes present in the sample as reference genomes, which would be the use-case in reality. All of these methods either produced high duplication ratios (low strain precision) or low strain recall (**Supplementary Table S1**).

Based on this, I believe that much more accurate and extensive comparison has to be done in the paper.

We believe that we made our comparison more accurate and extensive in the revised version. Thank you for your suggestions.

Minor comments:

- It is misleading that a plot on Figure S1 is called "histogram" several times.

Since the coverage values of a k -mer are an integer, the plot in Figure S2 (previously S1) actually is a histogram, the lines are just for visualisation purposes.

- References to the figures are probably not valid (for example, p6 l26 (Alg. 1, Fig. 2) -> (Alg. 1, Fig. 1); p10 l5: (Fig 3A) -> (Fig 2A); p15 l20: (Fig. 4) ??).

Thank you for indicating these issues. We have fixed the references accordingly.

- GitHub would benefit from a small working example.

That is a good suggestion. We accordingly have added a small fastq file to GitHub as a test and we uploaded all scripts to GitHub with the DOI <https://doi.org/10.5281/zenodo.4106497> and all data sets and additional material to publisso with the DOI.

References

1. Chen, J., Zhao, Y. & Sun, Y. De novo haplotype reconstruction in viral quasispecies using paired-end read guided path finding. *Bioinforma. Oxf. Engl.* **34**, 2927–2935 (2018).
2. Baaijens, J. A., Aabidine, A. Z. E., Rivals, E. & Schönhuth, A. De novo assembly of viral quasispecies using overlap graphs. *Genome Res.* **27**, 835–848 (2017).
3. Crits-Christoph, A. *et al.* Genome sequencing of sewage detects regionally prevalent SARS-CoV-2 variants. *medRxiv* 2020.09.13.20193805 (2020)
doi:10.1101/2020.09.13.20193805.
4. Sczyrba, A. *et al.* Critical Assessment of Metagenome Interpretation—a benchmark of metagenomics software. *Nat. Methods* **14**, 1063–1071 (2017).
5. Töpfer, A. *et al.* Probabilistic Inference of Viral Quasispecies Subject to Recombination. *J. Comput. Biol.* **20**, 113–123 (2013).
6. Hesse, U. *et al.* Virome Assembly and Annotation: A Surprise in the Namib Desert. *Front. Microbiol.* **8**, 13 (2017).
7. Li, D., Liu, C.-M., Luo, R., Sadakane, K. & Lam, T.-W. MEGAHIT: an ultra-fast single-node solution for large and complex metagenomics assembly via succinct de Bruijn graph. *Bioinforma. Oxf. Engl.* **31**, 1674–1676 (2015).
8. Nurk, S., Meleshko, D., Korobeynikov, A. & Pevzner, P. A. metaSPAdes: a new versatile metagenomic assembler. *Genome Res.* **27**, 824–834 (2017).
9. Olm, M. R. *et al.* InStrain enables population genomic analysis from metagenomic data and rigorous detection of identical microbial strains.
<http://biorxiv.org/lookup/doi/10.1101/2020.01.22.915579> (2020)
doi:10.1101/2020.01.22.915579.
10. Li, H. Minimap and miniasm: fast mapping and de novo assembly for noisy long sequences. *Bioinformatics* **32**, 2103–2110 (2016).
11. Shu, Y. & McCauley, J. GISAID: Global initiative on sharing all influenza data – from vision to reality. *Eurosurveillance* **22**, (2017).

12. Ke, Z. & Vikalo, H. A Graph Auto-Encoder for Haplotype Assembly and Viral Quasispecies Reconstruction. *Proc. AAAI Conf. Artif. Intell.* **34**, 719–726 (2020).
13. Fedonin, G. G., Fantin, Y. S., Favorov, A. V., Shipulin, G. A. & Neverov, A. D. VirGenA: a reference-based assembler for variable viral genomes. *Brief. Bioinform.* **20**, 15–25 (2017).
14. Ke, Z. & Vikalo, H. A Convolutional Auto-Encoder for Haplotype Assembly and Viral Quasispecies Reconstruction. *bioRxiv* 2020.09.29.318642 (2020)
doi:10.1101/2020.09.29.318642.
15. Chen, J., Shang, J., Wang, J. & Sun, Y. A binning tool to reconstruct viral haplotypes from assembled contigs. *BMC Bioinformatics* **20**, 544 (2019).
16. Hage, E. *et al.* Characterization of Human Cytomegalovirus Genome Diversity in Immunocompromised Hosts by Whole-Genome Sequencing Directly From Clinical Specimens. *J. Infect. Dis.* **215**, 1673–1683 (2017).
17. Sijmons, S., Van Ranst, M. & Maes, P. Genomic and Functional Characteristics of Human Cytomegalovirus Revealed by Next-Generation Sequencing. *Viruses* **6**, 1049–1072 (2014).
18. Simpson, J. T. *et al.* ABySS: A parallel assembler for short read sequence data. *Genome Res.* **19**, 1117–1123 (2009).
19. Peng, Y., Leung, H. C. M., Yiu, S. M. & Chin, F. Y. L. IDBA-UD: a de novo assembler for single-cell and metagenomic sequencing data with highly uneven depth. *Bioinforma. Oxf. Engl.* **28**, 1420–1428 (2012).
20. Boisvert, S., Laviolette, F. & Corbeil, J. Ray: Simultaneous Assembly of Reads from a Mix of High-Throughput Sequencing Technologies. *J. Comput. Biol.* **17**, 1519–1533 (2010).
21. Bankevich, A. *et al.* SPAdes: A New Genome Assembly Algorithm and Its Applications to Single-Cell Sequencing. *J. Comput. Biol.* **19**, 455–477 (2012).
22. Hunt, M. *et al.* IVA: accurate de novo assembly of RNA virus genomes. *Bioinforma. Oxf. Engl.* **31**, 2374–2376 (2015).

23. Yang, X. *et al.* De novo assembly of highly diverse viral populations. *BMC Genomics* **13**, 475 (2012).
24. Mikheenko, A., Saveliev, V. & Gurevich, A. MetaQUAST: evaluation of metagenome assemblies. *Bioinformatics* **32**, 1088–1090 (2016).
25. Deng, Z.-L. *et al.* Evaluating assembly and variant calling software for strain-resolved analysis of large DNA-viruses. *bioRxiv* 2020.05.14.095265 (2020)
doi:10.1101/2020.05.14.095265.
26. A reference standard for genome biology. *Nat. Biotechnol.* **36**, 1121–1121 (2018).
27. Splettstoesser, T. *English: Structure of the HIV-1 genome. It has a size of roughly 10.000 base pairs and consists of nine genes, some of which are overlapping.* (2014).
28. Mikheenko, A., Valin, G., Prjibelski, A., Saveliev, V. & Gurevich, A. Icarus: visualizer for de novo assembly evaluation. *Bioinformatics* **32**, 3321–3323 (2016).
29. Antipov, D., Korobeynikov, A., McLean, J. S. & Pevzner, P. A. hybridSPAdes: an algorithm for hybrid assembly of short and long reads. *Bioinforma. Oxf. Engl.* **32**, 1009–1015 (2016).
30. Wick, R. R., Judd, L. M., Gorrie, C. L. & Holt, K. E. Unicycler: Resolving bacterial genome assemblies from short and long sequencing reads. *PLOS Comput. Biol.* **13**, e1005595 (2017).
31. Kolmogorov, M. *et al.* metaFlye: scalable long-read metagenome assembly using repeat graphs. *Nat. Methods* **17**, 1103–1110 (2020).
32. Garrison, E. & Marth, G. Haplotype-based variant detection from short-read sequencing. *ArXiv12073907 Q-Bio* (2012).
33. Koren, S. *et al.* Canu: scalable and accurate long-read assembly via adaptive k-mer weighting and repeat separation. *Genome Res.* gr.215087.116 (2017)
doi:10.1101/gr.215087.116.
34. Truong, D. T., Tett, A., Pasolli, E., Huttenhower, C. & Segata, N. Microbial strain-level population structure and genetic diversity from metagenomes. *Genome Res.* **27**, 626–638 (2017).

35. Fritz, A. *et al.* CAMISIM: simulating metagenomes and microbial communities. *Microbiome* **7**, 17 (2019).
36. Compeau, P. E. C., Pevzner, P. A. & Tesler, G. Why are de Bruijn graphs useful for genome assembly? *Nat. Biotechnol.* **29**, 987–991 (2011).
37. Chikhi, R. & Medvedev, P. Informed and automated k-mer size selection for genome assembly. *Bioinformatics* **30**, 31–37 (2014).
38. Shariat, B., Movahedi, N. S., Chitsaz, H. & Boucher, C. HyDA-Vista: towards optimal guided selection of k-mer size for sequence assembly. *BMC Genomics* **15**, S9 (2014).
39. Baaijens, J. A., Stougie, L. & Schönhuth, A. Strain-aware assembly of genomes from mixed samples using flow variation graphs. *bioRxiv* 645721 (2020) doi:10.1101/645721.

Second round of review

Reviewer 1

The re-ordering and the additional explanation do well address the issues mentioned for the initial version of the manuscript.

Reviewer 2

The authors have adequately addressed my concerns. I noticed that the authors used "data set" and "dataset" in different places. Better be consistent about this.

Reviewer 3

Unfortunately, the authors do not address all my concerns. First of all, the formal description of the method is poorly written. For example:

1) Page 4. The graph the authors refer to as "deBruijn graph" is usually called an overlap graph, and there is a long discussion about the difference between the two in the assembly community (see, for example, Compeau, Phillip EC, Pavel A. Pevzner, and Glenn Tesler. "How to apply de Bruijn graphs to genome assembly." *Nature biotechnology* 29.11 (2011): 987-991.).

By the definition of the edges the author gave each edge (u,v) is presented in the graph together with its counterpart (v,u). It is not clear why this is the case, in particular, the unitig graph definition makes no sense with this definition of the "deBruijn" (actually, overlap) graph edges.

The graph on the figure S1 is definitely something different from what the authors defined on the page 4.

Since there is no formal proof of the algorithm correctness, there may be a number of errors which are critical for the method. For example:

2) Page 6: The authors claimed: "If genomes are very closely related, then the peaks will consist of k-mers that are unique to the individual strains and there will be another peak for the common k-mers." They also commented on this in the answer to the reviewer: "All k-mers for which two genomes differ will show in the histogram and produce a peak specific for the unique k-mers. Since these k-mers have to exist (otherwise the two strains would be the same), some peaks will show."

I believe this is an important assumption for the whole algorithm, and unfortunately it does not always hold. Assume for example, that two haplotypes are very similar, except the second one is just longer than the first one (so, the first one is its prefix). Then there are no unique k-mers of the first haplotype. Another important issue may be the result of recombination (which happens in viral genomes a lot): consider there are similar haplotypes I, II, III, and IV. For the sake of simplicity let's assume that they differ only in 3 positions, and there are more than k bp between them. Let me skip all the nucleotides except the different ones, and write the haplotype as I = -A-A-A-, II = -A-C-C-, III = -C-A-C-, IV = -C-C-A-. All of the haplotypes would have no unique k-mers (for example, I and II would share all the k-mers containing the first SNP position), but the haplotypes are still distinguishable (and even the haplotype calling problem may be solvable).

3) Page 6: "In the first step, the source vertex (with an in-degree of 0) with the highest coverage is selected from the unitig graph."

Such a vertex is not necessarily unique. It is not clear what would happen if there are several source vertices with the same highest coverage.

4) Page 7: "Additionally, putative paths can get removed, if too many of its edges are already part of a previous putative path (Supplementary Methods)."

The details are extremely important here, and I did not find the description of the putative paths in the submission.

Minor issues:

This is great to see the working example on github. The short description of the output may help the users. I was not able to find the strain abundances for the provided example.

Reviewer #1: The re-ordering and the additional explanation do well address the issues mentioned for the initial version of the manuscript.

Reviewer #2: The authors have adequately addressed my concerns. I noticed that the authors used "data set" and "dataset" in different places. Better be consistent about this.

Thank you for indicating this. We now consistently use the term „data set“.

Reviewer #3: Unfortunately, the authors do not address all my concerns. First of all, the formal description of the method is poorly written. For example:

*1) Page 4. The graph the authors refer to as "deBruijn graph" is usually called an overlap graph, and there is a long discussion about the difference between the two in the assembly community (see, for example, Compeau, Phillip EC, Pavel A. Pevzner, and Glenn Tesler. "How to apply de Bruijn graphs to genome assembly." *Nature biotechnology* 29.11 (2011): 987-991.).*

The definition that we included in the last manuscript version followed the definition of the review cited by reviewer 3, which we referenced exactly for this purpose. It is straightforward to distinguish between deBruijn graphs and overlap graphs. To clarify, the most distinctive aspect is that the former have k -mers as vertices and the latter *reads*. There is no discussion about the differences between them in the field.

The reason for reviewer 3's misunderstanding may have been that in the referenced review two different deBruijn graphs are depicted, namely in Figures 3c ($k=3$ analogous our original definition) and d ($k=2$ in our definition), while an example of an overlap graph is shown in Figure 3b, which is not entirely clear from the legend. Haploflow *clearly* uses a deBruijn graph, since it uses k -mers, not reads as vertices, and no overlaps between reads are calculated. **Importantly, the depicted edge labels in Fig 3 of the review are not part of the formal definition of deBruijn graphs.** To clarify this, we therefore refer the reader now to the definition in Pevzner et. al. (2001, 10.1073/pnas.171285098), which is analogous to the definition we gave for $k-1$.

Accordingly, we updated the definition in the manuscript **to the same as given by Pevzner et. al. 2001, p 9751, second paragraph**: "Given the reads $R = \{r_1, \dots, r_n\}$, a deBruijn graph $G = (V, E, k)$ contains all substrings of length $k-1$ of R as vertices V and two vertices u and v are connected with a directed edge if a substring of length k (called k -mer) exists which has u as prefix and v as suffix." (**page 4, third paragraph under "Results"**)

In case of any remaining doubts, we would also like to refer to Dr. Pevzner's group, who is the senior author on the above mentioned review and world-leading expert on this topic. Dr. Alexey Gurevich of Dr. Pevzner's group has reviewed our preprint and confirmed that our method is indeed based on a deBruijn graph. He would be happy to provide an expert

statement. We have furthermore consulted with a mathematician, who also confirmed this to be the case.

To summarize, we disagree with the claim that there is a discussion about the differences between overlap and deBruijn graphs, as they are straightforward to distinguish. We also strongly disagree with the claim that our method is based on an overlap graph, not a deBruijn graph. We would also like to point out that this is the first time the reviewer 3 made this claim, so we had no chance to address this with our previous revision.

By the definition of the edges the author gave each edge (u,v) is presented in the graph together with its counterpart (v,u) . It is not clear why this is the case, in particular, the unitig graph definition makes no sense with this definition of the "deBruijn" (actually, overlap) graph edges.

The graph on the figure S1 is definitely something different from what the authors defined on the page 4.

We do not think that the graph in Figure S1 is different from what was defined, and rephrased the formal definition in the manuscript to make it more clear (see above). If there remains a specific issue that is unclear, we are happy to incorporate or address this. Note that the edge description does not show the k -mer here, but a coverage value instead. Using the definition as outlined above, it is clear that Figure S1 (1) shows a deBruijn graph with $k = 5$ and Figure S1 (2) the corresponding unitig graph. We hope this has been addressed now using our explanations.

Since there is no formal proof of the algorithm correctness, there may be a number of errors which are critical for the method. For example:

To the best of our knowledge, it is not common nor required for assembly software used in metagenomics or viral genomics to be published together with a “formal proof of algorithm correctness”. To demonstrate this, we below provide a table for the state-of-the-art assembly techniques that we have included in our benchmarks below. None of these were published with a formal proof of algorithm correctness, nor was any other relevant paper that we found on the topic in Genome Biology. We therefore still think that it is outside of the scope of this journal and much less relevant for practical use on real data (which what this article is about) than the performance improvements across a range of data sets that we have shown. We do not wish to get into a large fundamental discussion about relevance of formal proofs, but considering all of the above, it seems to us an unreasonable request to provide a formal proof of algorithm correctness.

For a non-expert to be able to follow our arguments: one of the reasons that the above is the case is that oftentimes formal proofs are made on the basis of many simplifying assumptions, so their relevance for practical performance, in particular, for complex biological systems such as microbial communities, may have very limited value in practice - certainly less value

than the extensive demonstration of performance improvements on a range of data sets that we have provided.

Overview of formal proofs and formal descriptions of algorithms provided by scientific articles on state-of-the art assembly software:

SPAdes, J Comput Biol., 2012: no formal proof given. Relevant statement: “We assume that the reader is familiar with the concept of *A-Bruijn graphs* introduced in Pevzner et al.”

metaSPAdes, Genome Research, 2017: no formal proof, no formal description of DeBruijn graphs, relevant statement “*metaSPAdes* first constructs the de Bruijn graph of all reads using SPAdes, transforms it into the assembly graph using various graph simplification procedures, and reconstructs paths in the assembly graph that correspond to long genomic fragments within a metagenome”

MEGAHIT, Bioinformatics, 2015: no formal proof, no formal description of DeBruijn graphs, relevant statement “MEGAHIT makes use of succinct *de Bruijn* graphs (SDBG; *Bowe et al., 2012*), which are compressed representation of *de Bruijn* graphs”

PEHaplo, Bioinformatics, 2018: gives a proof of why their general algorithm is NP-complete and then - like the Haploflow manuscript - the pseudocode of a heuristic (greedy) version, but does not prove correctness in either case. In our case, this is not necessary, as the correctness of Haploflow’s fattest path algorithm follows *immediately* from the correctness of Dijkstra’s algorithm.

SAVAGE, Genome Research, 2017: no formal proof of correctness of their algorithm.

CliqueSNV, biorxiv, 2020: no formal proof given. Gives pseudo-code for part of the algorithms, but does not prove correctness of any parts. Moreover, the pseudocode often describes NP-complete algorithms, which are solved approximately (similar to PEHaplo).

QuasiRecomb, J Comput Biol., 2013: no formal proof given. Interestingly (relating to the point raised next by reviewer 3), the authors write in their publication: “[...]. The regularized model is not only computationally more convenient, but sparse recombination is also a biologically plausible assumption.”

MetaCarvel, Genome Biology, 2019: This is a scaffolding package published in *Genome Biology*. This manuscript does not give a single formal definition or proof of their algorithm, but instead relies on explaining the individual parts in natural language.

2) Page 6: The authors claimed: "If genomes are very closely related, then the peaks will consist of k-mers that are unique to the individual strains and there will be another peak for the common k-mers."

They also commented on this in the answer to the reviewer: "All k-mers for which two genomes differ will show in the histogram and produce a peak specific for the unique k-mers. Since these k-mers have to exist (otherwise the two strains would be the same), some peaks will show."

I believe this is an important assumption for the whole algorithm, and unfortunately it does not always hold.

This admittedly not entirely clear statement that we made in the rebuttal is **NOT an important assumption for the algorithm**. We are not sure why reviewer 3 is making that assumption, as this statement was not part of the manuscript, but of the rebuttal only. The k -mer histogram peaks are used to gain a priori strain abundance estimates that are used for error correction in preprocessing, but the actual strains assembled by the algorithm are based on the flow algorithm over the graph. Please also see our further explanations below for the examples given by reviewer 3.

Assume for example, that two haplotypes are very similar, except the second one is just longer than the first one (so, the first one is its prefix). Then there are no unique k -mers of the first haplotype. Another important issue may be the result of recombination (which happens in viral genomes a lot): consider there are similar haplotypes I, II, III, and IV. For the sake of simplicity let's assume that they differ only in 3 positions, and there are more than k bp between them. Let me skip all the nucleotides except the different ones, and write the haplotype as I = -A-A-A-, II = -A-C-C-, III = -C-A-C-, IV = -C-C-A-. All of the haplotypes would have no unique k -mers (for example, I and II would share all the k -mers containing the first SNP position), but the haplotypes are still distinguishable (and even the haplotype calling problem may be solvable).

First, the assumption that two strains differ in at least one k -mer is, biologically speaking, a very sound and also a common one. Similar assumptions are made in phylogenetics with the “infinite site assumption” (Kimura 1969; *Genetics* 61:893–903, Ma et. al. 2008, 10.1073/pnas.0805217105). Even in the case of recombination, it is highly unlikely that a genome does not have at least a single unique k -mer; indeed there is a software detecting recombination events by SNP incompatibility (Lai, YP., Ioerger, T.R. 2018, doi.org/10.1186/s12859-018-2456-z).

But even further, the cases described by the reviewer can be resolved by Haploflow quite well *exactly because* of its flow algorithm. If two strains are exactly equivalent, but one of them is a prefix of the other, then the *flow* will decrease after the last base of the shorter strain, Haploflow detects such changes in flow.

We write on **page 7, second paragraph**: “[...] coverages do not follow a normal distribution globally and consequently some consecutive edges in the assembly graph may exhibit steep changes in coverage” and further describe the reduction of flow on **page 34 and 35** (paragraph “*Algorithmic details of the flow algorithm*” in the supplement).

Figure R1: Schematic deBruijn graph for the three “haplotypes” $I = -A-A-A-$, $II = -A-C-C-$, $III = -C-A-C-$, $IV = -C-C-A-$ with abundances 60/30/7/3. Red paths indicate the chosen path from which the flow is reduced. Edges with 0 remaining flow are removed in the consecutive steps. The first path has coverage 63 and the sequence A-A-A (haplotype I). The second path has coverage 27 and the sequence A-C-C (haplotype II). The third path has coverage 6 and sequence C-C-C (erroneous). The last path has coverage 4 and sequence C-A-C (haplotype III).

In the second case, let us assume that the strains I-IV have abundances of - for example - 60, 30, 7 and 3. For the first difference the A- k -mer has a coverage of 90, and the C- k -mer a coverage of 10. At the second position, the A- k -mer has a coverage of 67 and the C- k -mer

one of 33. Finally, at the last position, the A-*k*-mer has a coverage of 63 and the C-*k*-mer one of 37. The first path would choose the highest abundance path, that is three times A (haplotype I). The coverage is assumed to be the coverage along the path, 63. The remaining coverages are 27/10 at the first junction, 4/33 at the second and 0/37 at the third. Now the second path follows the highest flow again, A-C-C (haplotype II). The highest flow is chosen as 27. The remaining coverages are 0/10, 4/6, 0/10. The next path is chosen as C-C-C (not present) with a coverage of 6. Finally, 0/4, 4/0, 0/4 are the final coverages and Haploflow will produce the contig with C-A-C (haplotype III).

Haploflow will a) correctly predict 4 haplotypes, b) predict 3 out of 4 haplotypes entirely correct and c) accurately predict coverages of 63 (+3), 27 (-3), 6 (-1) and 4 (+1), even though this example is a constructed one, which is highly unlikely to appear in such exact fashion biologically.

3) Page 6: "In the first step, the source vertex (with an in-degree of 0) with the highest coverage is selected from the unitig graph."

Such a vertex is not necessarily unique. It is not clear what would happen if there are several source vertices with the same highest coverage.

Again, it is possible to construct graphs for which two sources have exactly the same coverage, but in a biological context it is unlikely to happen. Even if it does happen, then Haploflow will choose the first one it encountered and start the algorithm from there. Probably the only change is the order of the contigs.

4) Page 7: "Additionally, putative paths can get removed, if too many of its edges are already part of a previous putative path (Supplementary Methods)."

The details are extremely important here, and I did not find the description of the putative paths in the submission.

On **page 34, first paragraph** under "Algorithmic details of the flow algorithm", in the Supplement we describe: "Since it is possible that edges are used multiple times, it is also possible that there are paths that have hardly any edges that are "unique" to that path. We call an edge unique, if it is part of exactly one path. If the fraction or length of unique edges of a path is too low, by default less than 500 bases, the path is removed for all edges on which it is not unique, to avoid overestimating the total number of paths in the graph". Generally, we disagree with the statement that these details are "extremely important", this is why we chose to add them only in the supplement and not in the main text.

Minor issues:

This is great to see the working example on github. The short description of the output may help the users. I was not able to find the strain abundances for the provided example.

This is a good suggestion. We have added a short description of the outputs. The strain abundances are stored in the name of the contig sequences in the fasta file (e.g. for the toy example the sequence names are “Contig_0_flow_1196.48_cc_0”, “Contig_1_flow_581.962_cc_0” and “Contig_2_flow_235.294_cc_0” for the three HIV strains. They are all in one connected component (cc_0) and they have flows of 1196, 581 and 235 (originally they were simulated as 10:5:2), Haploflows estimate is 10.17:4.94:2.

Third round of review

Reviewer 2

Usually it is quite difficult to have an algorithm that can be proved correct for metagenomic data analysis because there are too many complex factors in the real data. The design of the experiments and the results are more important to demonstrate the utility of the tools. Thus, I support the publication of this work.

The last sentence of the abstract "Haplotype reconstructed high-quality strain-resolved assemblies from clinical

HCMV samples and SARS-CoV-2 genomes from wastewater metagenomes identical to genomes from clinical isolates" is confusing and should be re-written.